# *CYP2C19* Genetic Variants and Major Depressive Disorder: A Systematic Review

**DOI:** 10.3390/ph17111461

**Published:** 2024-10-31

**Authors:** Larissa Sousa Silva Bonasser, Calliandra Maria de Souza Silva, Caroline Ferreira Fratelli, Bruna Rodrigues Gontijo, Juliana Moura Alves Seixas, Livia Cristina Lira de Sá Barreto, Izabel Cristina Rodrigues da Silva

**Affiliations:** 1Postgraduate Program in Health Sciences, University Campus Darcy Ribeiro, University of Brasília (UnB), Brasília-Federal District (DF), Brasília 70910-900, Brazil; laribonasser@gmail.com; 2Clinical Analysis Laboratory, Molecular Pathology Sector, Pharmacy Department, Faculty of Ceilândia, University of Brasília (UnB), Brasília-Federal District (DF), Brasília 72220-900, Brazil; cdssilva@gmail.com; 3Postgraduate Program in Health Sciences and Technologies, Faculty of Ceilândia, University of Brasília (UnB), Brasília-Federal District (DF), Brasília 72220-900, Brazil; carolfratelli@gmail.com (C.F.F.); brunargontijo.unb@gmail.com (B.R.G.); ju.seixas13@gmail.com (J.M.A.S.); 4Pharmacy Course, Faculty of Ceilândia, University of Brasília (UnB), Brasília-Federal District (DF), Brasília 72220-900, Brazil; liviabarretofarm@hotmail.com

**Keywords:** *CYP2C19*, depressive disorder, major, polymorphism, genetic

## Abstract

Major depressive disorder (MDD) affects over 300 million people globally and has a multifactorial etiology. The CYP2C19 enzyme, involved in metabolizing certain antidepressants, can influence treatment response. Following the PRISMA protocol and PECOS strategy, this systematic review assessed the variation in common *CYP2C19* gene variants’ frequencies across populations with MDD, evaluating their impact on clinical characteristics and treatment response. We comprehensively searched five databases, identifying 240 articles, of which only nine within the last decade met our inclusion criteria. Except for one study that achieved 74.28% of STROPS items, the rest met at least 75% of GRIPS and STROPS guidelines for quality and bias risk assessment. The *CYP2C19*’s *1 allele, the *1/*1 genotype, and the NM phenotype, considered as references, were generally more frequent. Other *CYP2C19* polymorphism frequencies exhibit significant variability across different populations. Some studies associated variants with MDD development, a more extended history of depression, prolonged depressive episodes, and symptom severity, while others reported no such association. Some studies confirmed variants’ effects on escitalopram and citalopram metabolism but not that of other drugs, such as sertraline, venlafaxine, and bupropion. Treatment tolerability and symptom improvement also varied between studies. Despite some common findings, inconsistencies highlight the need for further research to clarify the role of these polymorphisms in MDD and optimize treatment strategies.

## 1. Introduction

The term depression generally refers to any depressive disorder. Nonetheless, according to the DSM-V (*Diagnostic and Statistical Manual of Mental Disorders, Fifth Edition*), these disorders are divided into major depressive disorder (MDD, also known as major depression), persistent depressive disorder, disruptive mood dysregulation disorder, among others [1,2,3]. Three hundred and thirty-two million cases of depressive disorders were estimated worldwide in the year 2021, with a higher prevalence among people of the female biological sex in the age groups 15–19 and 60–64 years [4].

The etiology of MDD is believed to be multifactorial, including biological, genetic, environmental, and psychosocial factors [5]. The DSM-V defines depression as the presence of at least five symptoms that persist for at least two weeks, one of which is depressed mood or loss of interest or pleasure. Symptoms include depressed mood most of the day, markedly reduced interest or pleasure most of the day, weight loss or gain, decreased or increased appetite, insomnia or hypersomnia, agitation or psychomotor retardation, feelings of worthlessness or excessive guilt, fatigue or loss of energy, decreased ability to think or concentrate, recurrent thoughts of death, and suicidal ideation [2,5,6].

Several therapies are recommended for depression, which include psychological and psychosocial interventions, physical activities and treatments, as well as pharmacological treatments [6]. In terms of pharmacological treatment, antidepressants are considered the first-line treatment for moderate to severe depression; however, there is considerable individual variability in response to treatment. These differences in response may be related to several factors, such as drug interactions, subtypes of depression, comorbidities, smoking, and genetic variations, especially those found in genes responsible for metabolizing medications [7]. Approximately 15 to 30% of the variability in treatment response can be attributed to genetic variants that affect drug absorption, metabolism, transport, and mechanism of action [8].

Human cytochrome P450 (P450s or CYPs) proteins are a large superfamily of membrane-bound enzymes that contain heme as a cofactor. Each encoded by a different CYP gene, these enzymes are involved in more than 90% of the reported enzymatic reactions, catalyzing a wide range of organic substrates’ oxidative transformation, and their functions are pivotal in detoxifying xenobiotics (like drugs) and endogenous substrates, cellular metabolism, and homeostasis [9,10,11]. Six cytochrome P450 genes from the CYP1, 2, and 3 families encode enzymes responsible for metabolizing about 90% of clinical drugs [10,12,13]. These are *CYP1A1*, *CYP1A2*, *CYP2C9*, *CYP2C19*, *CYP2D6*, and *CYP3A4* [13], which can influence drug responses by affecting the drug’s pharmacological action, safety, and bioavailability. Genetic polymorphisms and epigenetic changes in the *CYP2B6*, *CYP2D6*, *CYP3A4*, and *CYP2C19* genes significantly influence the metabolism of antidepressant and antipsychotic drugs and may be responsible for interethnic and interindividual variations in drug therapeutic efficacy [10,11,14].

The *CYP2C19* gene contains nine exons that encode a protein with 490 amino acids. This gene is located on chromosome 10q23.33, along with other members of the CYP2C family, including *CYP2C8*, *CYP2C9*, *CYP2C18*, and *CYP2C19*. The CYP2C19 enzyme helps metabolize a range of clinically utilized medicines, as reported extensively in the literature [15,16,17,18], including approximately 15.5% of FDA-approved and routinely prescribed psychiatric drugs, such as citalopram and sertraline, as well as endogenous substances, such as melatonin and progesterone [8,18]. The enzyme’s functional classification divides individuals into five categories: poor metabolizers (PM), intermediate (IM), normal (NM), rapid (RM), and ultrarapid (UM) [18].

The *CYP2C19* gene is highly polymorphic, exhibiting a multitude of genetic variations that result in varying degrees of enzymatic activity, including complete absence, increased activity, or decreased activity [13]. Genetic variations in drug metabolism enzyme and transporter genes are typically reported as haplotypes—combinations of single-nucleotide polymorphisms (SNPs), insertion/deletion (InDels), copy number variants (CNVs), or other sequence variants [19,20]. The *CYP2C19* gene has over 35 functional haplotypes, often referred to as “star (*) alleles” for interpretation purposes. The “star allele” nomenclature standardizes genetic polymorphism annotations (pharmacogenomic markers) by assigning a unique star-allele identifier to known pharmacogenetic haplotypes or gene-level haplotype patterns that are typically associated with protein activity levels [16,19,20]. As a result, understanding the combination within a specific haplotype and an individual’s diploid content helps streamline research into drug metabolism, response, and adverse drug reactions [16,19,20]. Table 1 describes the *CYP2C19* gene’s most studied polymorphisms and their corresponding star-allele identifier in MDD. In contrast, Table 2 presents Table 1’s complementary haplotype phenotypes and their gene-level genetic variation description.

Many studies have sought to gain insight into how polymorphisms of the *CYP2C19* gene affect treatment outcomes, adverse effects, and their frequency in different populations, trying to identify gaps in this understanding [17,30,31,32]. These findings have led to the Pharmacogene Variation Consortium (PharmVar) catalog’s star (*) allele nomenclature for the polymorphic human *CYP2C19* gene and a reviewed consensus on pharmacogenomic testing and their effectiveness in psychiatry, explaining how *CYP2C19* genetic variation impacts the metabolism of many drugs and informing medication selection and dosing of several commonly used antidepressant and antipsychotic medications [16,18,33]. In light of the global prevalence of MDD, the variability in treatment response between individuals, and the CYP2C19 enzyme’s role in metabolizing antidepressant drugs, this systematic review aimed to determine the *CYP2C19* genetic variants’ frequency variation in different populations with major depressive disorder and to understand how these polymorphisms influence MDD clinical characteristics and the response to antidepressants.

## 2. Materials and Methods

### 2.1. Search Strategy and Selection Criteria

The present systematic review followed the Preferred Reporting Items for Systematic Reviews and Meta-Analyses (PRISMA) guidelines, PROSPERO number CRD42024525997. Its inclusion criteria were based on the Population, Exposure, Comparison, Outcome, and Study-type (PECOS) strategy, considering (1) population: human research participants with major depressive disorder; (2) exposure: *CYP2C19* genetic variants; (3) comparison: the *CYP2C19* variant’s genotype frequency; (4) outcome: *CYP2C19* variant’s genotype frequency fluctuation in different populations with MDD; (5) study type: observational and interventional.

For this, we included observational or interventional studies that presented data on the *CYP2C19* variant’s genotypic frequencies in human research participants with MDD and described their laboratory methods according to the eligibility criteria. However, studies with incomplete data (including statistical data), reviews, meta-analyses, abstracts, and studies not in English, Spanish, or Portuguese were excluded.

In March 2024, we conducted a comprehensive search using the databases EMBASE, Web of Science, PubMed (MEDLINE), Virtual Health Library (BVS), and the CAPES/MEC Journal Portal. Initially, no filters were applied, including the year of publication. However, due to the large volume of publications, the scope was later narrowed to articles published within the last decade to select more recent articles. The search terms used were “*CYP2C19* OR *CYPIIC19*”, “Depressive Disorder, Major”, and “Polymorphism, Genetic OR variant”, as defined by the Medical Subject Headings (MeSH) vocabulary thesaurus. These terms were combined using the Boolean operator “AND” and organized as follows: ((*CYP2C19* OR *CYPIIC19*) AND (Depressive Disorder, Major) AND (Polymorphism, Genetic OR variant)).

### 2.2. Study Selection and Data Extraction

Two reviewers (LB and CS) collaborated on the article selection in two phases. In the first phase, each reviewer independently analyzed each article’s title and abstract, verifying their eligibility according to the PECOS strategy. The Rayyan tool, developed by the Qatar Computing Research Institute (QCRI), assisted with this initial analysis and helped remove duplicates. In the second phase, the same two reviewers (LB and CS) also independently analyzed the full text of the preselected articles, consistently applying the pre-established eligibility criteria.

In both phases, disagreements or doubts were discussed between the two reviewers, and if an agreement could not be reached, a third reviewer (IS) was consulted. The two reviewers (LB and CS) then independently extracted predefined data into an electronic spreadsheet using Microsoft Office Excel. The extracted data included the following: author, study title, objective, year of publication, country of the study, diagnostic instrument for MDD, studied variants, *CYP2C19* variant’s genotype and phenotype frequencies, sample size, whether it was part of a larger trial or databank, laboratory methodology, main result, and *p*-value.

### 2.3. Bias Risk in Each Study

Genetic risk models are typically based on examining genetic variants or analyzing genetic and environmental risk factors to predict disease risk, prognostic outcome, treatment response, or treatment-related harms [34]. Hence, we initially, given the scope of our systematic review, employed 20 (26 considering 4 items’ subitems) of the Genetic Risk Prediction Studies (GRIPS) guideline 25 items that evaluated the article’s methods (7 items plus 8 subitems), results (6 items plus 2 subitems), and discussion (3 items), to assess the risk of bias in the selected studies and verify their quality and completeness [34], whereas an article was classified as of good quality if it presented at least 75% (of 15 to 26 applicable items, depending on the article, including subitems) of the evaluated items.

Nevertheless, as many of the observational and interventional studies selected were pharmacogenetic studies, we also assessed the articles by Strengthening the Reporting of Pharmacogenetic Studies (STROPS) Guideline, which includes 54 items and their subitems developed to improve the pharmacogenetic studies reporting transparency and to facilitate the conduct of high-quality systematic reviews and meta-analyses [35]. This analysis considered 49 (56 considering 2 items’ subitems) STROPS items that evaluate the article’s methodology (27 items plus 6 subitems), results (12 items plus 3 subitems), discussion (4), and other relevant information (4), checking for their presence or absence. Articles were deemed good quality if they met at least 75% (of 35 to 56 applicable items, depending on the article, including subitems) of these items.

To avoid complications that could affect the bias risk analysis, two reviewers (LB and CS) independently conducted both these assessments for all the selected articles, and any disagreements were resolved after discussion with the third reviewer (IS). Nonapplicable items were disregarded when calculating the total number of guideline items selected for each article. From this value, to arrive at the final percentage, the total of applicable items, including subitems, was subtracted from the criteria not met.

## 3. Results

### 3.1. Article Search, Selection, and Quality Assessment

Initially, 240 articles were identified across the four databases searched. After removing duplicates, 172 studies remained for the title and abstract analysis, taking into account the aspects defined in our PECOS strategy. Of these, 45 articles persisted for full-text review. Finally, after applying the inclusion and exclusion criteria, 9 articles were included in this systematic review (Figure 1, Table 3). Our choice of the limited publication period (2014–2024) favored the selection of more recent information on the research subject. As the fields of pharmacology, genetic screening, and gene–disease modeling are constantly updated, this approach gives insights into the latest publications. The excluded articles and their reason for exclusion are described in Appendix A.

### 3.2. General Characteristics of the Selected Studies

As shown in Figure 2, most studies were conducted on the American (United States, Canada, and Trinidad and Tobago) and European (Germany, Serbia, and Poland) continents, followed by transcontinental Turkey and Asia (China). Regarding demographic data, most research was conducted with adults over 18, and women were more prevalent in the major depressive disorder (MDD) groups.

### 3.3. CYP2C19 Variants Star Genotypic and Allelic Frequency

Among individuals with MDD, the *1/*1 (wild-type) genotype generally had the highest frequency, followed by the *1/*17 and *1/*2 genotypes (Figure 3, Appendix A). The *1/*3 and *3/*3 genotypes were rare or found in few or no patients (Figure 3). Of the alleles, *1 was the most common (ranging from 41% to 86%), followed by *2 (ranging from 11% to 41%) and *17 (ranging from 7% to 30%) (Figure 4, Appendix A). The least common was *3, which ranged from 0% to 1% (Figure 4, Appendix A). This systematic review did not consider the uncommon variants rs28399504 (*4), rs1853205, rs4986894, and rs12767583 [21,22].

### 3.4. CYP2C19 Phenotypic Frequency

In terms of phenotypes, the normal metabolizer—NM phenotype had the highest frequency in the populations studied (18% to 75%), followed by the intermediate metabolizer—IM (18% to 46%) and rapid metabolizer—RM (6% to 32%) phenotypes, as seen in Figure 5. The poor metabolizer (PM) and ultrarapid metabolizer (UM) were the least frequent phenotype (Figure 5, Appendix A). The combined phenotypes, slow metabolizer—SM (PM and IM) and fast metabolizer—FM (RM and UM), were similar in most studies (Figure 5). The exceptions were Montané et al. [38] (Trinidad and Tobago), Świechowski et al. [39] (Poland), and Zhang et al. [41] (China) studies in which the SM phenotype was more prominent (Figure 5, Appendix A).

### 3.5. Analysis of Bias Risk in Each Study

Appendix A present the selected articles’ bias risk analysis and quality determination results using the Genetic Risk Prediction Studies (GRIPS) [20 items of 25 items] and the Strengthening the Reporting of Pharmacogenetic Studies (STROPS) [49 items of 54 items] guidelines, respectively. Of the articles analyzed by GRIPS, 100% (9) had at least 75% (of 15 to 26 applicable items, depending on the article, including subitems) or more items and were considered of good quality [25,36,37,38,39,40,41,42,45], see Appendix A. The lowest score was 93.3% (14 items) [25]. On the other hand, of those analyzed by STROPS, 88.9% (8) had at least 75% (of 35 to 50 applicable items, depending on the article, including subitems) or more items and were considered of good quality [36,37,38,39,40,41,42,45], while the lowest score was 74.28% (26 applicable items and subitems) [25], see Appendix A.

## 4. Discussion

### 4.1. CYP2C19 Polymorphisms and Their Genotypic and Phenotypic Frequencies in Major Depressive Disorder

Most studies in European, transcontinental Turkey (Asia/Europe), and American (North and Caribbean) populations found a higher NM (normal metabolizer) phenotype (*1/*1 genotype) frequency among major depressive disorder (MDD) patients, with IM (intermediate metabolizer) and RM (rapid metabolizer) alternating as the second most frequent and PM consistently being the least common (Figure 5, Table 3 and Appendix A). In contrast, the IM phenotype was most prevalent in Asian populations, followed by NM, and the PM phenotype was the least prevalent (Figure 5, Table 3 and Appendix A).

In the European population, the highest phenotype frequencies found by Hahn et al. [25] (Germany) were 32% for the NM and 30% for RM. While for the Joković et al. [40] (Serbia) study, the two highest frequencies were also for NM (40.20%) and RM (30.39%) phenotypes. Considering that both studies evaluated the *1 and *17 polymorphisms, this trend in phenotypic frequencies makes sense. In contrast, the two highest frequencies found in the Świechowski et al. [39] (Poland) study were for NM (74.8%) and IM (22.3%). One reason for this difference might be that Świechowski et al. [39] did not evaluate the *17 polymorphism and its associated RM phenotype, which would explain why the second most common phenotype was IM. Despite this difference, the least common phenotype in all studies was PM (Figure 5, Table 3 and Appendix A).

Regarding the NM phenotype, this review’s findings corroborate those found in other studies with European populations. Sim et al. [47] reported a higher *1/*1 (NM) genotype frequency in a Swedish population treated with antidepressants, in which of 1416 patients, 613 had the *1/*1 genotype (43.3%). Calleja et al. [48], in a clinical trial with 98 healthy Spanish individuals, also found that the most common phenotype was NM (57.1%). Finally, Joas et al. [49] also found a higher frequency of the NM phenotype (characterized in this study as EM, extensive metabolizer, another definition for NM) in a study of 5019 Swedish patients diagnosed with bipolar disorder, where 2187 (43.6%) patients had this phenotype.

In the studies from the American continent, the NM phenotype was the most common in the individuals evaluated. Islam et al. [42] (Canada) observed a 40.11% frequency for this phenotype, while the second most common phenotype was IM (28.81%). Kharasch et al. [45] (United States) also found the most frequent to be the NM phenotype (38.81%). In contrast, their second most frequent phenotype was RM (31.34%). Interestingly, these studies’ least frequent phenotypes diverged, with the Canadian being PM (2.83%) and the United States being UM (2.99%). These disparities may be due to the polymorphisms associated with poor metabolization assessed and to the size of the samples, as shown in Figure 5 and Table 3, Appendix A and Appendix A. On the other hand, the Caribbean country Trinidad and Tobago presented different degrees of frequencies depending on their self-reported Afro- or Indo-Trinidadian origin [38]. In both subgroups, the most common phenotype was IM (45%; 46%), followed by NM (40%; 17.95%), while the least common PM (5%; 20%) and RM (10%; 15%) varied between the groups, respectively (Figure 5 and Table 3 and Appendix A). The same can be observed regarding their genotypic and allelic frequencies (Figure 5, Table 3 and Appendix A).

Still on the American continent, Veldic et al. [50] analyzed 1795 patients diagnosed with MDD and bipolar disorder (BP), finding that 3.5% of the patients exhibited the PM phenotype, 27.4% the IM phenotype, and 69.1% the NM phenotype. They also found a higher PM phenotype frequency in BP patients than in those with MDD (9.3% vs. 1.7%). The most frequent genotypes in this study were *1/*1 (69.09%) and *1/*2 (25.55%). These results differ in part from those described by Collins et al. [51], who evaluated 75 American (United States) patients with various mood disorders and found that NM was the most common phenotype (30.7%), followed by RM (25.3%). In a separate study involving 227 patients with various psychiatric disorders, Hall-Flavin et al. [52] found that, in both the pharmacogenetic test-guided and nonguided treatment groups, NM was the most frequent phenotype (75% and 72%, respectively), followed by IM (22.2% and 26.9%, respectively). While the findings of these studies align with the most common phenotypes observed in this review, variations in the second most common phenotype were noted. These discrepancies may be influenced by the number of individuals analyzed and the specific polymorphisms studied.

In transcontinental populations of Turkey (Asia/Europe), Uckun et al. [36] and Yuce-Artun et al. [37] found similar frequencies for the NM phenotype, which was the most common in both groups (48% and 44%, respectively). However, these studies’ second most common phenotype differed (RM and IM, respectively) (Figure 5, Table 3 and Appendix A). Although the two studies had the same total number of participants (50), differing only in the number of males and females, the number was small, which may have confounded these differences.

Regarding transcontinental populations that, like Turkey, span the same Asia and European continents, Zastrozhin et al. [53] observed that the most frequent phenotype in individuals was NM, detected in 64.6% of patients, followed by the IM phenotype (35.4%) in a study of 130 Russian male patients diagnosed with depressive episodes, mental disorders, and behavioral disorders associated with alcohol use. Their most frequent phenotype aligns with those described in this review. The second most common phenotype results were only partially consistent, as shown in Figure 5 and Table 3 and Appendix A. These differences again show that frequencies can differ in populations from the same continent.

Zhang et al.’s study [41] of a Chinese population found that the IM phenotype was the most common (44.42%), followed by the NM phenotype (42.60%) (Figure 5, Table 3 and Appendix A). This finding aligns with the results reported by Kim et al. [54], who analyzed a Korean population of 13,160 patients undergoing percutaneous coronary intervention and treated with dual antiplatelet therapy. In a subgroup of 2266 individuals genotyped for *CYP2C19* polymorphisms, the most common phenotype was also IM (47.97%), followed by NM (38.22%). Similarly, Xi et al. [55] studied 41,090 Chinese patients who underwent percutaneous coronary intervention and were treated with dual antiplatelet therapy, finding a higher frequency of the IM phenotype (50.1%) compared to NM (35.8%). However, the absence of *17 polymorphism analysis in Xi et al.’s study [55] limits the ability to make a more detailed comparison with the findings of this review, regardless of the difference in the commodity studied.

### 4.2. CYP2C19 Polymorphisms and MDD’s Clinical Characteristics

Regarding the studied populations’ clinical characteristics, Świechowski et al. [39] found no association between the *CYP2C19*2* polymorphisms and MDD severity before pharmacotherapy or the disorder’s onset time (Table 3). However, Zhang et al. [41] demonstrated that *CYP2C19**3 A allele carriers were 2178 times more likely to develop MDD than noncarriers. Furthermore, the A-G haplotype (rs4986893-rs4244285) correlated with an increased risk in developing MDD (OR = 2.306, *p* = 0.001), suggesting that the *CYP2C19* gene may also be associated with susceptibility to developing MDD beyond its role in antidepressant drug metabolism [41]. Joković et al. [40] further observed that patients with the SM (PM and IM) phenotype had a significantly longer history of depression and a longer duration of the current depressive episode compared to the NM and FM (RM and UM) groups. Moreover, the SM group had a significantly higher baseline score on the 21-item revised Beck Depression Inventory (BDI-IA) scale compared to the FM group, reinforcing the association between the *CYP2C19* gene and MDD patients’ clinical characteristics.

Similar to Świechowski et al. [39], Athreya et al. [56], conducting a predictive analysis with 1030 MDD patients of different ethnicities treated with citalopram and escitalopram, found that the *CYP2C19* gene metabolization phenotypes were also not associated with these individuals’ clinical characteristics, such as severity of depression, nor their demographic characteristics. Likewise, Morinobu et al. [28] observed no significant differences in onset age, HAMD scale scores before treatment, or imipramine dosage between Japanese MDD patients with and without the *CYP2C19′*s *2 (m1) and *3 (m2) polymorphisms. In contrast, Kanders et al. [57], in a study with 150 Swiss MDD patients, found that the *CYP2C19**17 CC genotype carriers had higher scores on the CES-D scale (Center for Epidemiologic Studies Depression Scale) compared to the *17 CT and TT genotypes, implying a possible association with MDD severity. Inversely, Sim et al. [47], when analyzing the *2 and *17 polymorphisms in 1416 Europeans treated with antidepressants, found that the *2/*2 (PM) genotype had a significantly lower CES-D scale score and, consequently, fewer depressive symptoms compared to the *1/*1 (NM) reference genotype. Notably, when stratified by biological sex, this association persisted only in the male group.

Concerning the concentrations of antidepressants and their metabolites, Uckun et al. [36] observed that the *CYP2C19* *1/*1 genotype carriers had higher plasma desmethylcitalopram (DCIT) concentrations compared to the *CYP2C19**2 allele carriers, confirming this polymorphism influence on citalopram (CIT) metabolism. However, these two groups did not present significant differences in their CIT concentrations. In contrast, Yuce-Artun et al. [37] found no differences in the mean concentrations of sertraline (SERT) and desmethylsertraline (DSERT) across different *CYP2C19* polymorphisms’ genotypes. Similarly, Montané et al. [38] observed no contribution of *CYP2C19* polymorphisms to the venlafaxine/O-desmethyl venlafaxine metabolic ratio, with patients carrying the *CYP2C19* *17/*17 (UM) genotype showing similar results to those with the PM genotype. Kharasch et al. [45] also found no effect of *CYP2C19* phenotypes on bupropion plasma concentrations, bupropion hydroxylation, or hydroxybupropion stereoisomer concentrations. However, they noted that these phenotypes did affect other metabolic pathways, such as bupropion ketogenic reduction and 4′-hydroxylation. These findings confirm that while *CYP2C19* polymorphisms may affect the metabolism of certain antidepressants, they may not affect others [15,16,17,18]. Nevertheless, in order to clarify this point, it is crucial to evaluate confounding factors that may interfere with this metabolization since they generally cannot be controlled entirely [17], such as the patient’s consumption of grapefruit juice [58].

When analyzing *CYP2C19* polymorphisms’ influence in a European population, Huezo-Diaz et al. [29] found that compared to the *1/*1 (NM) genotype carriers, *17 allele (associated with RM and UM phenotypes, see Table 2) carriers had a significantly lower escitalopram serum concentration. Conversely, carriers of the PM phenotype (*2/*2, *2/*3, and *3/*3 genotypes, see Table 2) exhibited significantly higher escitalopram concentrations. On the other hand, they observed no significant differences in N-desmethylescitalopram concentration between the groups, contrary to the CIT concentration results reported by Uckun et al. [36]. Supporting Huezo-Diaz et al.’s findings [29], when analyzing the *2 polymorphism in a Russian population, Zastrozhin et al. [59] observed that GA (IM) genotype carriers had a higher CIT serum concentration compared to GG (NM) genotype carriers. Unfortunately, a comparison with the AA (PM) genotype was impossible due to the absence of this genotype in the sample. In line with the results of Huezo-Diaz et al. [29] and Zastrozhin et al. [59], Tsai et al. [27] evaluated the *CYP2D6*, *CYP2C19*, and *CYP3A4* genes’ impact on escitalopram plasma concentration and treatment response in 100 Asian patients. After analyzing the *CYP2C19* *2, *3, and *17 polymorphisms, they found that MDD patients with the PM phenotype (*2/*2, *2/*3, and *3/*3) had significantly higher escitalopram serum concentrations compared to those with the IM (*1/*2 and *1/*3) and NM (*1/*1) phenotypes.

Not all of the selected studies assessed the *CYP2C19* gene polymorphisms’ association with MDD clinical characteristics, revealing heterogeneity in regard to which outcomes each study evaluated. The ones that did, analyzed the possibility of these polymorphisms associated with MDD development, as well as with a more extended history of depression, longer duration of the current depressive episode, and greater symptom severity [40,41]. Some confirmed these associations [57], while others reported no such association [28,39,47,56].

The plasma concentrations of the antidepressant drugs and their metabolites were also not evaluated in all selected articles. Uckun et al., for example, found that the *CYP2C19*2* polymorphism impacted desmethylcitalopram plasma concentrations [36]. Other studies agree with these results, showing the impact of *CYP2C19* gene polymorphisms on the metabolization of escitalopram and citalopram [27,29,59]. Alternately, other of the selected studies found no association between *CYP2C19* gene polymorphisms and the sertraline/desmetilsertraline, venlafaxine/O-desmetilvenlafaxine, and bupropion metabolization, as demonstrated by their unchanged plasma levels [37,38,45].

These dissimilarities among the selected studies demonstrate inconsistencies regarding the *CYP2C19* gene polymorphisms’ association with MDD clinical characteristics and certain drug metabolizations, making it essential for studies to analyze confounding factors to confirm associations in varying contexts. Following study-type guidelines could also help standardize methods to regulate variables and compare results, thus improving the understanding of factors affecting the study’s outcomes.

### 4.3. CYP2C19 Polymorphisms and Treatment Response and Tolerance

Regarding the response to treatment in the studies reviewed, Świechowski et al. [39] found a statistically significant correlation between the *CYP2C19**2 allele presence and clinical improvement after pharmacotherapy, as assessed by the Hamilton Depression Rating Scale (HDRS), as carriers of at least one *2 allele exhibited better responses to treatment. Similarly, Joković et al. [40] observed that although all evaluated genotypes reduced HAMD and BDI-IA scale scores during follow-up, patients with the SM (IM or PM) phenotypes experienced a less pronounced reduction than those with the NM phenotype. Furthermore, the SM phenotypes group had a notably lower prevalence of treatment response than the NM group.

In studies examining adverse reactions and side effects, Islam et al. [42] found that patients in the SM phenotype groups (PM and IM, see Table 2 for *CYP2C19* phenotypes and corresponding genotypes) treated with escitalopram alone experienced more significant treatment-related changes in sexual arousal compared to those treated with a combination of escitalopram and aripiprazole. Overall, the SM group showed improvements in sexual arousal, while the FM (RM and UM) group did not differ significantly from the NM group. In contrast, Joković et al. [40] reported that the SM group had a significantly higher side-effects score than the NM group, particularly with regard to neurological side effects such as nervousness and restlessness, as well as a higher mean gastrointestinal symptoms score. Notably, contrary to Islam et al.’s findings [42], Joković et al. [40] observed no differences in sexual function between the SM and NM groups. Additionally, the SM group had lower scores on the Clinical Global Impression–Efficacy (CGI-E) scale compared to the NM group.

Other authors have found similar results to those reported by Świechowski et al. [29] and Joković et al. [40]. For instance, Zastrozhin et al. [53] evaluated 267 male individuals with depressive episodes and alcohol-related mental and behavioral disorders. After 4 and 8 weeks of follow-up, they found that the HADS and HAMD scale scores decreased in all genotype groups for the *CYP2C19**17 polymorphism. However, the wild-type genotype (CC, see Table 2 for possible combinations) group experienced a more pronounced increase in the UKU Side-Effect Rating Scale score. In another study of 227 patients with depressive disorders, Hall-Flavin et al. [52] analyzed various *CYP2C19* polymorphisms (*1, *2, *3, *4, *5, *6, *7, and *8) and evaluated differences between patients with treatment guided and not guided by pharmacogenetic tests. They found that patients with pharmacogenetically guided treatment showed a significant reduction in depressive symptoms on the HAMD-17 scale compared to the nonguided group. Arnone et al. [33] reported similar results in a meta-analysis, finding that when pharmacogenomic tests were employed to tailor the treatment of depression, they were more effective than the usual methods according to the improvement assessed in MDD patients.

Calabrò et al. [26] studied patients from the GSRD study group treated with at least one of several antidepressants, including amitriptyline, citalopram, clomipramine, doxepin, escitalopram, imipramine, sertraline, and trimipramine. Their results indicated that patients with the PM phenotype were more likely to respond to these treatments and less likely to show resistance than those with the NM phenotype. Furthermore, the PM phenotype group exhibited greater symptom improvement and experienced more autonomic and neurological adverse reactions than the other phenotype groups.

Contrary to the findings of Świechowski et al. [39] and Joković et al. [40], Schosser et al. [60] reviewed studies on patients with treatment-resistant depression and found no significant association between their metabolic profiles linked to the *CYP2C19* gene and treatment response or remission in both the GSRD and the STAR*D study samples. Based on these findings, the authors concluded that metabolic profiling might not reliably predict antidepressant response or remission rates but could be valuable for anticipating side effects from drug interactions. Similarly, Zastrozhin et al. [59] analyzed the *CYP2C19**2 polymorphism and observed that although HAM-D and HADS scores decreased by the end of the follow-up period, the GA genotype carriers still had higher scores than noncarriers, indicating less improvement in those with the A allele. Furthermore, the A allele carriers also showed higher UKU scores, suggesting a greater susceptibility to adverse effects in this group.

In line with these findings, Serretti et al. [61] analyzed European patients with MDD treated with different antidepressant classes and observed that metabolic profiles associated with the *CYP2C19**2 and *17 polymorphisms showed no significant association with treatment response or symptom remission. Likewise, Morinobu et al. [28] found no significant difference in the rate of improvement in treatment and the severity of side effects between patients with and without the *CYP2C19**2 (m1) and *3 (m2) polymorphisms in a group of Japanese MDD patients treated with imipramine. Further supporting these observations, Tsai et al. [27] evaluated the *CYP2C19**2, *3, and *17 polymorphisms in Asian patients and found no differences in treatment response, as estimated by HAM-D and HAM-A scale scores, between the PM and SM groups. Lastly, Ng et al. [62] assessed *CYP2C19* polymorphisms in Australian and Asian patients taking escitalopram and venlafaxine and found, in the escitalopram-treated group, no statistically significant differences in HDRS scale score reduction when comparing the PM/IM and NM/UM phenotypes. However, they did observe that the NM and UM groups had increased autonomic side effects compared to the PM and IM groups.

Finally, Peters et al. [63] studied the STAR*D cohort to determine whether pharmacokinetic genes, including *CYP2C19* *2, *3, and *17 polymorphisms, could affect the response or tolerability to citalopram in patients with nonpsychotic MDD. The study found no significant association between the PM phenotype and response or tolerability to citalopram compared to the NM phenotype. Similarly, Taranu et al. [64] examined 182 MDD Caucasian patients treated with venlafaxine from the METADAP study, evaluating *CYP2C19* *2, *3, *4, *5, and *17 polymorphisms. Their results indicated no significant differences between the phenotype groups in treatment response, as measured by the HDRS scale.

Regarding the response to treatment, among the selected studies that evaluated this outcome, patients with the SM phenotype also improved their conditions after pharmacotherapy [39,40], even though those with the NM phenotype had a more pronounced improvement [40]. This relationship has been confirmed by several authors [26,33,52,53]. However, contrasting results have also been found, showing no association between *CYP2C19* gene polymorphisms and a better response to treatment [27,28,59,60,62,64].

Concerning treatment tolerability, determined by the occurrence of side effects and adverse reactions, among the reviewed results, MDD patients with the SM phenotype had more adverse reactions and side effects than patients with the NM phenotype [40,42]. Other studies confirmed [59] or found no differences [63] regarding the lower tolerability to treatment in the SM group. The same was observed in terms of sexual function improvement following therapy, with one study finding no difference in improvement between the phenotype groups [40] and another indicating that the SM phenotype groups improved more than the other groups [42].

These dissimilarities in results reiterate the importance of standardizing treatment and following study-type guidelines to regulate and better understand the elements that can generate disparate results of different studies.

### 4.4. Quality Assessment and Limitations of Selected Articles

Most of the studies included in this systematic review are pharmacogenetic studies. It was, therefore, crucial to apply not only the GRIPS guideline (Appendix A), which is widely used to assess the quality of genetic risk prediction studies but also STROPS (Appendix A), which is more suitable for pharmacogenetic studies. In this way, it was possible to assess the quality of the studies with greater precision and clarity, providing more transparent results.

All of the selected articles described the participants’ eligibility criteria, except for one [25], in which the criteria were not applicable (database). Additionally, they all discussed the results, generalizing and demonstrating the relevance of their findings; however, two studies did not address their limitations [25,37]. Specifically, with regard to STROPS, only one study clearly reported how it carried out their samples’ genotyping quality [39], and only one demonstrated how the sample calculation was conducted [40].

All these findings show that the quality of the articles included in this review was generally adequate. However, particular areas for improvement can be identified for application in future studies, which would promote more transparency for the results and enable replication and generalization of the findings by other researchers.

## 5. Conclusions

This systematic review reveals notable variations in the genotypic and phenotypic frequencies of *CYP2C19* gene polymorphisms among MDD patients of different populations, even within the same continent. In Europe, the NM (normal metabolizer) phenotype was the most common, with PM (poor metabolizer) being the least common. In the Americas, the predominant phenotype varied between NM and IM (intermediate metabolizer), with the least common ranging from PM to RM (rapid metabolizer) and UM (ultrarapid metabolizer). In Asia, IM was the most frequent, and PM was the least. Furthermore, in Turkey, a transcontinental region, NM was the most common, and PM was the least common. This variability likely reflects differences in the specific polymorphisms analyzed, sample sizes (often small), and the distinct ethnic and cultural/environmental factors of the populations studied.

Clinical features also displayed some variability. A study identified an association between the *CYP2C19**3 A allele and MDD development. Among studies assessing MDD severity, one found no association between the *CYP2C19**2 polymorphism and MDD baseline severity before pharmacotherapy. In contrast, another noted that the SM group had higher baseline BDI-IA scores than the FM group. Metabolic assessment parameters also showed inconsistencies. Some studies reported no significant differences in serum concentrations of antidepressants and their metabolites—such as sertraline, desmethylsertraline, venlafaxine/O-desmethylvenlafaxine, and bupropion—across phenotypes. However, one study found that the *CYP2C19* *1/*1 (NM) group had higher desmethylcitalopram plasma concentrations compared to *CYP2C19**2 allele carriers, though no differences were observed in citalopram concentrations between the two groups. These findings suggest that *CYP2C19* polymorphisms may influence the metabolism of some antidepressants but not others, highlighting the need to consider other factors that could interfere with metabolism.

Treatment response results were more consistent. One study demonstrated a significant correlation between the *CYP2C19**2 allele presence and clinical improvement following pharmacotherapy. At the same time, another found that all phenotype groups reduced HAMD and BDI-IA scores over time. However, the SM phenotype exhibited a less pronounced reduction compared to the NM phenotype. Adverse reactions and side effects also varied. One study reported that PM and IM groups treated with escitalopram alone experienced more significant changes in sexual arousal compared to those treated with both escitalopram and aripiprazole, while another found no differences in sexual function between groups. The latter study also found that the SM group had significantly higher side-effect scores and lower CGI-E scale scores than the NM group.

This review had some limitations. Although all the studies reported allelic, genotypic, and phenotypic frequencies—our primary focus—the populations studied varied in particular characteristics and the number of participants; for instance, some had a small sample size. These differing respective characteristics and sample sizes might limit the generalizability of this review’s results. Furthermore, there was also heterogeneity in the outcomes assessed, where some investigated the plasma concentrations of drugs and their metabolites, for example, and others investigated the response and tolerability to treatment. Lastly, not all studies analyzed the same *CYP2C19* polymorphisms, making it hard to comprehensively analyze all the phenotypes and genotypes.

Overall, genotyping for *CYP2C19* polymorphisms holds significant potential for tailoring MDD treatment, as this gene plays a crucial role in metabolizing various antidepressants. However, inconsistencies in the literature underscore the need for further research into these polymorphisms. Standardization in selecting polymorphisms to screen, larger sample sizes, and greater ethnic diversity in studies could yield more representative and accurate results. Moreover, controlling for additional factors influencing treatment outcomes is essential for achieving more reliable findings.

## Figures and Tables

**Figure 1 pharmaceuticals-17-01461-f001:**
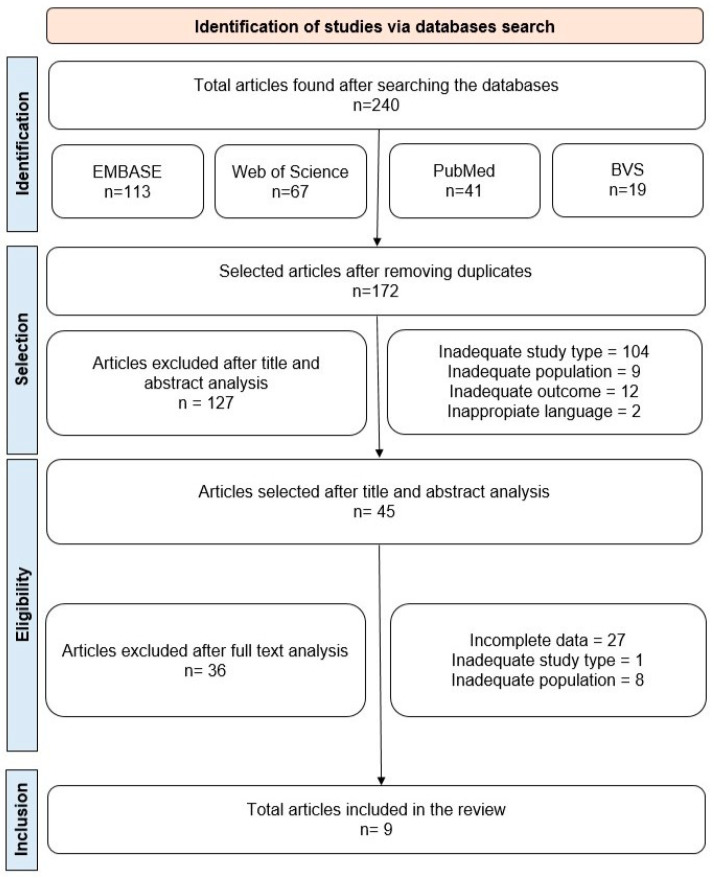
Flowchart outlining the steps adopted in the selection of articles.

**Figure 2 pharmaceuticals-17-01461-f002:**
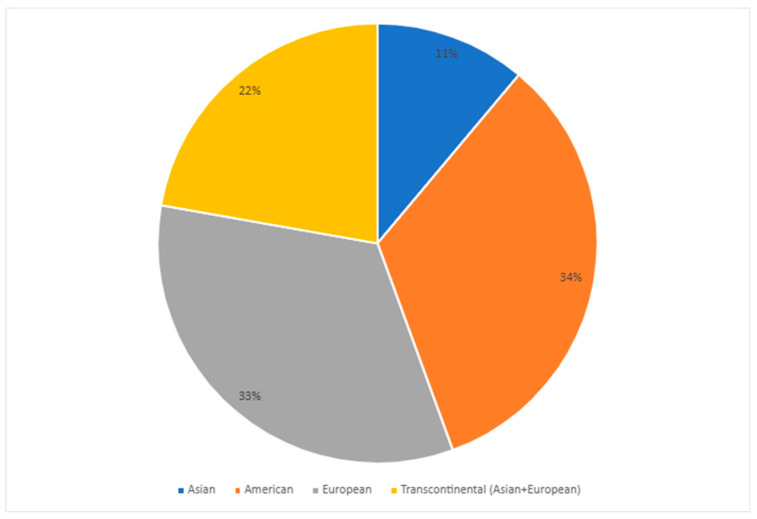
Distribution of articles published in the last decade on the most common *CYP2C19* gene polymorphisms in populations with major depressive disorder (MDD) per continent [25,36,37,38,39,40,41,42,45].

**Figure 3 pharmaceuticals-17-01461-f003:**
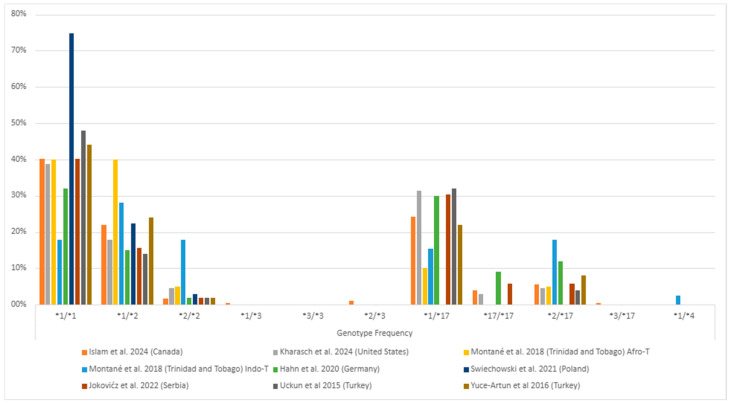
The most common *CYP2C19* gene polymorphisms’ genotypic frequency in the studied populations with major depressive disorder (MDD) [25,36,37,38,39,40,42,45]. Zhang et al. [41] (China) is missing as they only informed their gene-level genetic variants description. The * followed by a number provides the star allele identifier for standardized genetic variations (haplotype) that affect drug responses.

**Figure 4 pharmaceuticals-17-01461-f004:**
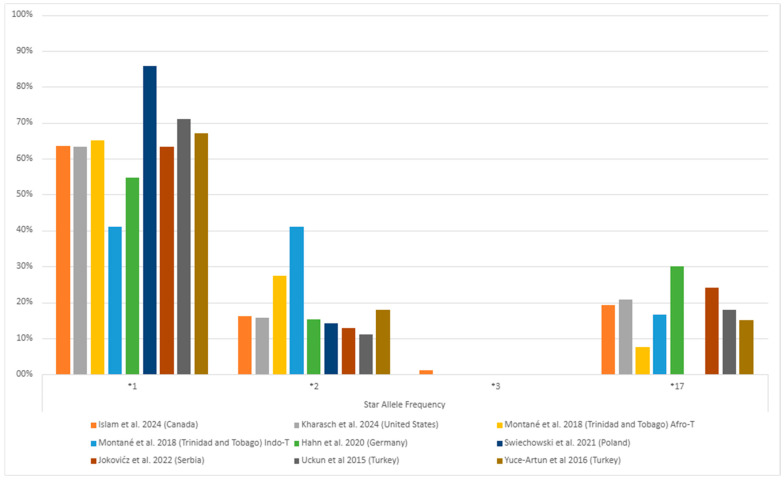
The most common *CYP2C19* gene polymorphisms’ alleles frequency in the studied populations with major depressive disorder (MDD) [25,36,37,38,39,40,42,45]. Zhang et al. [41] (China) is missing as they only informed their gene-level genetic variants description. The * followed by a number provides the star allele identifier for standardized genetic variations (haplotype) that affect drug responses.

**Figure 5 pharmaceuticals-17-01461-f005:**
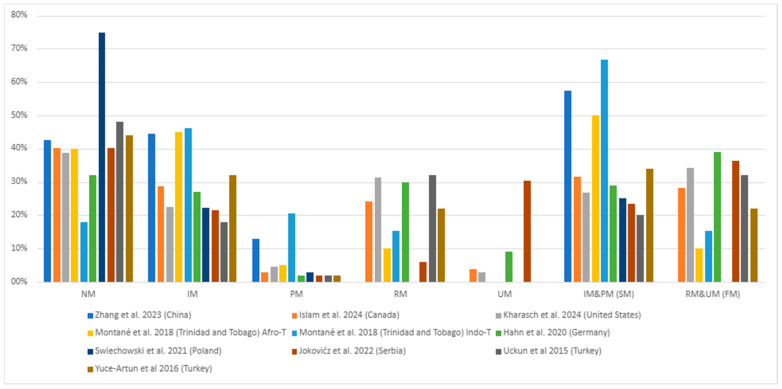
The most common *CYP2C19* gene polymorphisms’ phenotypic frequency in the studied populations with major depressive disorder (MDD) [25,36,37,38,39,40,41,42,45]. IM = Intermediate metabolizer; NM = Normal metabolizer; PM = Poor metabolizer; SM = Slow metabolizer (PM and IM); RM = Rapid metabolizer; UM = Ultrarapid metabolizer; FM = Fast metabolizer (RM and UM).

**Table 1 pharmaceuticals-17-01461-t001:** *CYP2C19* gene’s most studied polymorphisms with their star-allele identifier, haplotype description, and their product-associated enzymatic activity [21,22,23,24,25,26,27,28].

Reference SNP Cluster ID (rsID)	Star-Allele Identifier	Haplotype Description	Reference Base	Alternate Base	Location/Type of Mutation	Enzymatic Activity	Variation Consequence
wild-type reference (wt; *1)	*1	Wild-type, G allele at rs4244285, G allele at rs4986893 and C allele at rs12248560	G/C	-	-	Normal	-
rs4244285 (c.681G>A, p.Pro227Pro; *2; m1)	*2	681G>A, A allele at rs4244285 and G allele at rs4986893 (exon 5)	G	A	Splice-site mutation	Inactive	Loss of function
rs4986893 (c.636G>A; *3; m2)	*3	636G>A, G allele at rs4244285 and A allele at rs4986893 (exon 4)	G	A	W212X	Inactive	Loss of function
rs12248560 (c.-806C>T; *17)	*17	-806C>T, T allele at rs12248560	C	T	SNP in the promoter region	Increased	Gain of function

Note: The * followed by a number provides the star allele identifier for standardized genetic variations (haplotype) that affect drug responses, e.g., *CYP2C19*1*, where *1 is a normal function allele (the reference allele) of the gene *CYP2C19*. G—guanine, C—cytosine, A—adenine, and T—thymine, wt—wild-type, m1—*CYP2C19**2, m2—*CYP2C19**3, Pro—Proline, W—Tryptophane, X—Termination, p.—protein, c.—coding DNA sequence.

**Table 2 pharmaceuticals-17-01461-t002:** *CYP2C19* gene’s most studied haplotype phenotypes and their gene-level genetic variants description, haplotype pattern, and star-allelic combination [26,29].

Haplotype Phenotype	Genotype Description	Genotype			Star-Allelic Combination *
		rs4244285	rs4986893	rs12248560	
Normal metabolism (NM)/ Extensive metabolism (EM)	Alleles without functional change	GG	GG	CC	*1/*1
Intermediate metabolism (IM)	1 nonfunctional allele	GA	GG	CC	*1/*2
Poor metabolism (PM)	2 nonfunctional alleles	AA	GG	CC	*2/*2
Intermediate metabolism (IM)	1 nonfunctional allele	GG	GA	CC	*1/*3
Poor metabolism (PM)	2 nonfunctional alleles	GG	AA	CC	*3/*3
Poor metabolism (PM)	2 nonfunctional alleles	GA	GA	CC	*2/*3
Rapid metabolism (RM)	1 allele with increased function	GG	GG	CT	*1/*17
Ultrafast metabolism (UM)	2 alleles with increased function	GG	GG	TT	*17/*17
Intermediate metabolism (IM)	1 nonfunctional allele	GA	GG	CT	*2/*17
Intermediate metabolism (IM)	1 nonfunctional allele	GG	GA	CT	*3/*17

Note: * Except for NM’s *1/*1 genotype, the haplotype phenotype can have other more uncommon star-allelic combinations that maintain the genotype/functional description. For instance, the *4 (rs28399504) allele produces an inactive enzyme; therefore, carriers have either IM or PM phenotypes depending on their allelic combination with other polymorphisms. G—guanine, C—cytosine, A—adenine, and T—thymine.

**Table 3 pharmaceuticals-17-01461-t003:** Comparison of studies published in the last decade investigating the most common *CYP2C19* gene polymorphisms in populations with major depressive disorder (MDD).

Author	Year	Title	Country	Objective	Instrument	Sample	*CYP2C19* Genetic Variant	*CYP2C19* Star Genotype Frequency	*CYP2C19* Phenotype Frequency	Laboratory Test	Part of a Bigger Trial/Databank	Results for *CYP2C19* Genetic Variants	*p*-Value
Uckun et al. [36]	2015	The impact of *CYP2C19* polymorphisms on citalopram metabolism in patients with major depressive disorder	Turkey	Determine *CYP2C19* genetic polymorphisms and evaluate their impact on CIT metabolism in a Turkish population sample.	SCID-I	**MDD =** 50 (7 males and 43 females) **HC =** 209 (120 males and 89 females)	*1 (wt) *2 (rs4244285) *17 (rs12248560)	**MDD** *1/*1 = 24 (48%) *1/*2 = 7 (14%) *2/*2 = 1 (2%) *1/*17 = 16 (32%) *2/*17 = 2 (4%) **HC** *1/*1 = 139 (66.5%) *1/*17 = 61 (29.2%) *17/*17 = 9 (4.3%)	**MDD** NM = 24 (48%) IM = 9 (18%) PM = 1 (2%) RM = 16 (32%) **HC** NM = 139 (66.5%) RM = 61 (29.2%) UM = 9 (4.3%)	PCR-RFLP	No	The *1/*1 (wt) group and the other genotype subgroups did not significantly differ regarding age, CIT dose, and body weight. The mean plasma DCIT concentrations had a pronounced difference between *1/*1 and *1/*2 + *2/*2 genotype group. The mean C/D (dose-corrected) plasma DCIT levels were also significantly higher in the *1/*1 group than in the *1/*2 + *2/*2 groups. Moreover, the mean metabolic ratio (MR: CIT-to-DCIT) was also significantly higher in the *1/*2 + *2/*2 group than in the *1/*1 group. No differences between *1/*1 and *1/*17+*2/*17 subjects regarding CIT and DCIT, MR, and C/D of DCIT plasma concentrations were found. Similarly, plasma CIT concentrations were no different between *1/*1 and *1/*2 + *2/*2 carriers. However, DCIT plasma concentrations and C/D values were significantly higher in the *1/*1 group than in the *1/*2 + *2/*2 group. The MR value was significantly higher in the *1/*2 + *2/*2 group than in the *1/*1 group.	*1/*1 vs. *1/*2+*2/*2 for DCIT plasma concentration: *p* < 0.05 *1/*2+*2/*2 vs. *1/*1 for C/D plasma DCIT levels: *p* < 0.05 *1/*1 vs. *1/*2 + *2/*2 for MR, CIT/DCIT: *p* < 0.05
Yuce-Artun et al. [37]	2016	Influence of *CYP2B6* and *CYP2C19* polymorphisms on sertraline metabolism in major depression patients	Turkey	Evaluate the *CYP2B6* *6 and *9 polymorphisms’ influence on the steady-state SERT and DSERT plasma concentrations in MDD patients receiving SERT treatment and investigate the effects of *CYP2C19* *2 and *17 polymorphisms among the study group.	SCID-I	**MDD = 50** (15 males and 35 females)	*1 (wt) *2 (rs4244285) *17 (rs12248560)	**MDD** *1*1 = 22 (44%) *1*2/ = 12 (24%) *2/*2 = 1 (2%) *1/*17 = 11 (22%) *2/*17 = 4 (8%)	**MDD** NM = 22 (44%) IM = 16 (32%) PM = 1 (2%) RM = 11 (22%)	PCR–RFLP	No	Mean SERT and DSERT plasma concentrations, normalized SERT (concentration/dose ratio: determined concentration divided by SERT daily dose, ng/mL/mg), normalized DSERT, and DSERT/SERT values did not differ significantly among the four *CYP2C19* genotype subgroups.	-
Montané et al. * [38]	2018	Impact of *CYP2D6* on venlafaxine metabolism in Trinidadian patients with major depressive disorder	Trinidad and Tobago	Assess *CYP2D6* and *CYP2C19* variants’ impact on VEN at steady-state in MDD patients of Indian and African descent from Trinidad and Tobago.	DSM-IV criteria and the HAMD17 score	**MDD =** 59 * (Afro-T = 20 and Indo-T = 39) (19 males (33.3%) and 38 females (66.7%))	*1 (wt) *2 (rs4244285) *3 (rs4986893) *4 (rs28399504) *17 (rs12248560)	**MDD** Afro-T *1/*1 = 8 (40%) *1/17 = 2 (10%) *1/*2 = 8 (40%) *2/*17 = 1 (5%) *2/*2 = 1 (5%) Indo-T *1/*1 = 7 (17.95%) *1/17 = 6 (15.38%) *1/*2 = 11 (28.21%) *1/*4 = 1 (2.56%) *2/*17 = 7 (17.95%) *2/*2 = 7 (17.95%)	**MDD** Afro-T NM = 8 (40%) IM = 9 (45%) PM = 1 (5%) RM = 2 (10%) Indo-T NM = *1/*1 = 7 (17.95%) IM = 18 (46.15%) PM = 8 (20.51%) RM = 6 (15.38%)	TaqMan assays	No	*CYP2C19* variants did not contribute to the VEN/ODV metabolic ratio. UM (*17/*17) carriers had a similar range and mean metabolic ratio of VEN/ODV as PMs carrying two no-function alleles (*2/*2).	-
Hahn et al. [25]	2021	Frequencies of genetic polymorphisms of clinically relevant gene-drug pairs in a German psychiatric inpatient population	Germany	Analyze *CYP2D6* and *CYP2C19* genetic polymorphisms (and others) frequencies in psychiatric patients with depressive episodes admitted as inpatients at the Vitos Klinik Eichberg.	ICD-10: (F33.2, F33.3, F32.2, F34.1, and F33.1)	**MDD** = 108 (46 males and 62 females) (43% males and 57% females)	*1 (wt) *2 (rs4244285) *3 (rs4986893) *4 (rs28399504) *17 (rs12248560)	*1/*17 (RM) = 32 (30%) *1/*1 (NM) = 35 (32%) *1/*2 (IM) = 16 (15%) *2/*2 (PM) = 2 (2%) *17/*17 (UM) = 10 (9%) *2/*17 (IM) = 13 (12%)	NM = 35 (32%) IM = 29 (27%) PM = 2 (2%) RM = 32 (30%) UM = 10 (9%) (31% heterozygous = new rapid metabolizer definition)	Genetic testing kit by Humatrix AG (Pfungstad, Germany)	No	*CYP2C19* polymorphisms were present in 73 (68.6%) patients. Forty-two patients (38.9%) were UMs and RMs with a high risk of not responding to *CYP2C19* substrates (e.g., CIT, ESC, some tricyclics) at the prescriber’s information recommended dosage. Notably, only 14 patients (13%) were NM for *CYP2C19* and NM for *CYP2D6*.	UM = 9% (CI 0.3629–0.5551) RM = 31% (CI 0.2228–0.3972) NM = 32% (CI 0.232–0.408) IM = 27% (CI 0.1863–0.3537) PM = 2% (CI –0.0064–0.0464)
Świechowski et al. [39]	2021	The influence of *CYP2C19**2 and *CYP3A5**3 variants on the development of depression and effectiveness of therapy: A preliminary study	Poland	Determine the *CYP3A5**3 and *CYP2C19**2 alleles frequency in MDD patients and healthy controls to identify any association with MDD development and progression and the effectiveness of pharmacotherapy.	ICD-10 (F33.0–F33.8); HDRS	**MDD** = 103 (34 males and 69 females) (33% males and 67% females) **HC** = 93 (34 males and 59 females) (37% males and 63% females)	*1 (wt) *2 (rs4244285)	**MDD** *1/*1 (NM) = 77 (74.8%) *1/*2 (IM) = 23 (22.3%) *2/*2 (PM) = 3 (2.9%) *1 =177 (85.9%) *2 =29 (14.1%) **HC** *1/*1 (NM) = 63 (67.7%) *1/*2 (IM) = 29 (31.2%) *2/*2 (PM) = 1 (1.1%) *1 =155 (83.3%) *2 = 31 (16.7%)	**MDD** NM = 77 (74.8%) IM = 23 (22.3%) PM = 3 (2.9%) **HC** NM = 63 (67.7%) IM = 29 (31.2%) PM = 1 (1.1%)	PCR-RFLP	No	The *2 polymorphism demonstrated no statistically significant association with the age of MDD onset nor with the severity of its symptoms before pharmacotherapy based on the HDRS I score. Furthermore, although the treatment effectiveness, calculated as the change in Hamilton score (HDRS I–HDRS II), presented no significant differences with *2 polymorphism (*p* = 0.1904), *2 allele correlated significantly with clinical condition improvement after pharmacotherapy (HDRS I–HDRS II), as patients with at least one *2 allele achieved signifi cantly better treatment results (*p* = 0.0239).	*CYP2C19**2 vs. age of MDD onset: *p* = 0.5067 *CYP2C19**2 vs. severity before pharmaco therapy: *p* = 0.7180 *CYP2C19**2 vs. better treatment results: *p* = 0.0239 MDD patients vs. HC genotype and allele frequencies: *p* = 0.4771
Joković et al. [40]	2022	*CYP2C19* slow metabolizer phenotype is associated with lower antidepressant efficacy and tolerability	Serbia	Determine whether the *CYP2C19* genotype is associated with changes in antidepressant ef ficacy and tolerability in MDD inpatients.	ICD-10; HDRS; Mini International Neuropsychiatric Interview 5.0.0; HDRS/HAMD; BDI-IA; TSES.	**MDD** = 102 (58 males and 44 females) (57% males and 43% females)	*1 (wt) *2 (rs4244285) *17 (rs12248560)	*1/*1 (NM) = 41 (40.20%) *2/*2 (PM) = 2 (1.961%) *1/*2 (IM) = 16 (15.69%) *2/*17 (IM) = 6 (5.882%) *17/*17 (UM) = 6 (5.882%) *1/*17 (RM) = 31 (30.39%)	**MDD** NM = 41 (40.20%) IM = 22 (21.57%) PM = 2 (1.961%) UM = 6 (5.882%) RM = 31 (30.39%) IM and PM (SM) = 24 (23.53%) RM and UM (FM) = 37 (36.27%)	TaqMan SNP Genotyping assays (Applied Biosystems, Foster City, CA, USA).	No	SMs exhibited a less pronounced HAMD and BDI-IA scores reduction and a lower treatment response rate than NMs. SMs also experienced higher treatment side-effects TSES intensity scores for the central nervous system and gastrointestinal adverse reactions compared to NMs, with a strong correlation between CNS-related adverse reactions and illness severity at hospital admission measured with baseline HAMD score, a moderate correlation between gastrointestinal and sexual function, adverse reaction intensity and symptom severity. The CGI-E score was significantly lower in the SM patients compared with NMs in both V1 and V2, with a 0.6 point amplitude difference on average in both visits. Compared with NM and RM, lower antidepressant efficacy and tolerability were observed in SMs. NMs and FMs presented no significant differences in these parameters, treatment outcomes, and trajectories. SMs were slightly older than FMs and had significantly longer histories of depression compared to NMs and FMs. SMs also had a significantly higher baseline BDI-IA score than FMs. There were no group-specific differences in mean intensity score of SF adverse reactions.	**SM** ↓ reduction in HDRS (HAMD) scores = *p* < 0.0001 ↓ reduction in BDI-IA scores = *p* < 0.0001 ↓ treatment response rate = *p* < 0.0001 longer histories of depression = *p* < 0.05
Zhang et al. [41]	2023	*CYP2C19*-rs4986893 confers risk to major depressive disorder and bipolar disorder in the Han Chinese population whereas ABCB1-rs1045642 acts as a protective factor	China	Investigate previously reported candidate genes’ polymorphisms for MDD and bipolar disorder (BPD) in the Han Chinese population.	DSM-V	**MDD** = 439 (158 males and 281 females) (36% males and 64% females) **HC** = 464 (196 males and 268 females) (42% males and 58% females)	*1 (wt) *2 (rs4244285) *3 (rs4986893) *17 (rs12248560)	**MAF** MDD *3(A) = 0.0547 HC *3(A) = 0.0259 ***CYP2C19**17 (rs12248560):** MDD CC = 434 (98.86%) MDD CT = 5 (1.139%) MDD TT = 0 (0.0%) HC CC = 460 (99.14%) HC CT = 4 (0.862%) HC TT = 0 (0.0%) ***CYP2C19**3 (rs4986893):** MDD GG = 393 (89.52%) MDD AG = 44 (10.02%) MDD AA = 2 (0.456%) HC GG = 440 (94.83%) HC AG = 24 (5.17%) HC AA = 0 (0.0%) ***CYP2C19**2 (rs4244285):** MDD GG = 219 (49.89%) MDD AG = 179 (40.77%) MDD AA = 41 (9.34%) HC GG = 256 (55.17%) HC AG = 168 (36.21%) HC AA = 40 (8.62%)	**MDD** NM = 187 (42.60%) IM = 195 (44.42%) PM = 57 (12.98%) IM and PM (SM) = 252 (57.40%) **HC** NM = 248 (53.45%) IM = 160 (34.48%) PM = 56 (12.07%) IM and PM (SM) = 216 (46.55%)	Shanghai Kangli Medical Research Institute assisted with SNP Genotyping using MassARRAY SpectroCHIP and MALDI-TOF mass spectrometer.	No	The *3 MAF in the MDD group (0.0547) was higher than the control group (0.0259, *p* < 0.05), leading to the *3 A allele having a 2.178 odds ratio (OR) for MDD. The impaired *CYP2C19* metabolism caused by the *3A-*2G haplotypes might confer the risk of MDD (χ^2^ = 11.145, OR = 2.306, *p* = 0.001). IM and PM frequencies were higher in MDD (57.40%, OR =1.547) cases than in controls (46.55%, *p* < 0.05).	*CYP2C19**3 Χ2 = 9.781; *p* = 0.002; OR = 2.178 *CYP2C19**2 Χ2 = 2.009; *p* = 0.156; OR = 1.16 *CYP2C19**17 Χ2 = 0.174; *p* = 0.676; OR = 1.323
Islam et al. # [42]	2024	Influence of *CYP2C19*, *CYP2D6*, and *ABCB1* gene variants and serum levels of escitalopram and aripiprazole on treatment-emergent sexual dysfunction: a Canadian Biomarker Integration Network in Depression 1 (CAN-BIND 1) Study	Canada	Investigate the *CYP2C19*, *CYP2D6*, and *ABCB1* gene polymorphisms’ association with treatment-emergent changes in sexual function and sexual satisfaction in the Canadian Biomarker Integration Network in Depression 1 (CAN-BIND-1) sample.	DSM-IV-TR and MADRS, and confirmed with the Mini-International Neuropsychiatric Interview.	**MDD** = 178 # (68 males and 110 females) (38.2% males and 61.8% females)	*1 (wt) *2 (rs4244285) *3 (rs4986893) *17 (rs12248560)	*1/*1 (NM) = 71 (33.65%) *1/*2 (IM) = 39 (18.48%) *2/*17 (IM) = 10 (4.739%) *1/*3 (IM) = 1 (0.474%) *3/*17 (IM) = 1 (0.474%) *2/*2 (PM) = 3 (1.421%) *2/*3 (PM) = 2 (0.948%) *1/*17 (RM) = 43 (20.38%) *17/*17 (UM) = 7 (3.318%)	NM = 71 (39.89%) IM = 51 (28.65%) PM = 5 (2.809%) RM = 43 (24.16%) UM = 7 (3.933%) Not known = 1 (0.57%)	-	Yes. CAN-BIND-1 study (ClinicalTrials.gov identifier: NCT01655706), see [43,44]	The *CYP2C19* metabolizer groups significantly associated with the treatment-related change in sexual arousal in the ESC-Only treatment arm following multiple testing corrections, F(2,54) = 8.00, *p* < 0.001, q = 0.048, whereas ESC + ARI did not demonstrate this association. Repeated measures linear mixed-effects analysis revealed that SMs improved sexual arousal from weeks 8–16 (B = 0.44, 95% CI: 0.21 to 0.67), while FMs did not differ significantly from NMs. *CYP2C19* metabolizer phenotypes may be influencing changes in sexual arousal related to ESC monotherapy, in which the NM (*1/*1) phenotype could be at an increased risk of experiencing SSRI-associated decline in sexual arousal.	*p* < 0.001 0.326
Kharasch et al. [45]	2024	Pharmacogenetic influence on stereoselective steady-state disposition of bupropion	United States	Evaluate *CYP2B6*, *CYP2C19*, and *P450 oxidoreductase* genetic polymorphisms’ influence on the disposition of Valeant Pharmaceuticals Wellbutrin brand bupropion in MDD participants.	MADRS	**MDD** = 67 (53 female (76%) and 14 male (24%)	*1 (wt) *2 (rs4244285) *17 (rs12248560)	*1/*1 (NM) = 26 (38.81%) *1/*2 (IM) = 12 (17.91%) *2/*2 (PM) = 3 (4.478%) *1/*17 (RM) = 21 (31.34%) *17/*17 (UM) = 2 (2.985%) *2/*17 (IM) = 3 (4.478%)	NM = 26 (38.81%) IM = 15 (22.39%) PM = 3 (4.48%) RM = 21 (31.34%) UM = 2 (2.99%)	All genotyping was performed by the Washington University in St. Louis Genome Technology Access Center.	Yes. Clinical Study of Generic and Brand Bupropion in Depression—BALANCE (ClinicalTrials.gov identifier: NCT02209597), see [46]	*CYP2C19* polymorphisms did not influence bupropion plasma concentrations or hydroxybupropion formation but did influence the minor pathway of 4′-hydroxylation of bupropion and primary metabolites.	*p* < 0.05 vs. wild-type

Note: MDD = Major depressive disorder; HC = Healthy controls; IM = Intermediate metabolizer; NM = Normal metabolizer; PM = Poor metabolizer; SM = Slow metabolizer (PM and IM); RM = Rapid metabolizer; UM = Ultrarapid metabolizer; FM = Fast metabolizer (RM and UM); SD = Standard error; CI = Confidence interval; Diagnostic and Statistical Manual of Mental Disorders, Fourth Edition (DSM-IV); Diagnostic and Statistical Manual of Mental Disorders, Fifth Edition (DSM-V); International Classification of Diseases, Tenth Revision (ICD-10); Hamilton Depression Rating Scale (HDRS/HAMD); 21-item revised Beck Depression Inventory (BDI-IA); Clinical Global Impression—Efficacy index (CGI-E); Toronto Side-Effects Scale (TSES); Structured Clinical Interview for DSM-IV Axis I Disorders (SCID-I); Montgomery–Åsberg Depression Rating Scale (MADRS), Selective serotonin reuptake inhibitors (SSRIs); Wild-type (wt); Citalopram (CIT); Demethylcitalopram (DCIT); Metabolic ratio (MR); Sertraline (SERT); Desmethylsertraline (DSERT); Escitalopram (ESC); Aripiprazole (ARI); Venlafaxine (VEN); O-desmethylvenlafaxine (ODV); MAF = Minor allele frequency; ORs = Odds ratios; Polymerase chain reaction-Restriction fragment length polymorphism (PCR-RFLP); Central Nervous System (CNS). * There is a discrepancy regarding the number of patients. The article cites 57 in the methods and 59 in the results. # There is a discrepancy regarding the number of patients. The article cites 178, but one was removed from further analysis because the phenotype was unknown.

## Data Availability

Data is contained within the article and Appendix A.

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
