# Peer review of "CYP2C19 Genetic Variants and Major Depressive Disorder: A Systematic Review"

_pharmaceuticals, 2024, doi:10.3390/ph17111461_

Round 1

Reviewer 1 Report

Comments and Suggestions for Authors

 CYP2C19 genetic variants and Major Depressive Disorder: a systematic review

The article “CYP2C19 genetic variants and Major Depressive Disorder: a systematic review“ explores and important issue of CYP 2C19 genetic variants and treatment response in MDD. While the topic is certainly worth exploring there are some significant limitations to the methodology of the study.

Abstract:  The abstract in generally well-structured and comprehensive. The reporting of review outcomes is very vague. You have found some data indicating that CYP2C19 polymorphisms are linked to some antidepressants plasma concentrations and treatment response in MDD, report that in the abstract.

Key words: there is an error which is repeated in many times in the manuscript, with the widely used term “Major Depressive Disorder” being written in an uncommon manner “Depressive Disorder, Major”. This need to be corrected throughout the text.

Introduction:

The first paragraph is inconsistent in naming some but not all diagnoses listed in the “Depressive Disorders” category of DSM-5. I see no need to list all types of depressive disorders in this part of the text, especially given that the article refers largely to MDD. Instead, I suggest replacing this part of the text with the later fragment explaining the term “Major depressive disorder” by listing its diagnostic criteria. This particular fragment needs to be rewritten with caution, as some symptoms are not properly described (i.e. “reduced physical movement” instead ofPsychomotor changes” or “sense of uselessness” instead of “sense of worthlessness”). If none of the authors works clinically, I believe it would be best if You would consult a clinician (psychiatrist) prior to providing revisions to the text.

The authors state that “The CYP2C19 enzyme helps metabolize a range of clinically utilized medicines, including approximately 15.5% of FDA-approved and routinely prescribed psychiatric drugs, such as citalopram and sertraline, as well as endogenous substances, such as melatonin and progesterone [6,13]”. This work has limitations but it can be of relevance to clinicians. Provide a list or a table of antidepressants which are metabolized by CYP2C19 to a potentially significant extent. Here are potential sources for such a table, but feel free to use any others if You have more comprehensive ones (doi: 10.1080/17425255.2024.2352468; 10.1001/jamapsychiatry.2020.3643).

Results:

Table 1 is well prepared, yet the abbreviations used in the table should be explained in the footnotes. Please provide them.

Table 2 is well prepared, yet the abbreviations used in the table should be explained in the footnotes. Please provide them.

The flow chart is comprehensible and well organized.

The description “American” comes up several times but it is not precise. Please specify whether You mean North/South American or both by “American”.

All figures are of very low quality. Please provide Figures with higher resolution.

In the section “2.2” the abbreviation “MDD” is all of a sudden introduced while it should ne provided in the part of the text where the phrase “Major Depressive Disorder” appears first.

Table 3 has misspells and lacks appropriate use of MDD abbreviations, there are some hyperlinks in the Table instead of a proper citation. Please correct this table once more and do it thoroughly. Regarding Table 3, why do the authors report the assessed drugs for some studies but do not for others? Whenever the data is available the drugs evaluated in a particular study should be listed in the table.

Discussion:

Please specify whether You mean North/South American or both by “American”.

The “BDI-IA” needs to be explained upon it’s first occurrence in the text.

If You wish to use abbreviation for the word “citalopram” introduce it upon it’s first occurrence in the text, same goes for “sertraline”, “desmethylsertraline” etc.

The discussion is quite long but it is a description of the results rather than a discussion. Discuss Your results with the relevant literature i.e. 10.1001/jamapsychiatry.2020.3643, 10.1111/bcp.15762 , 10.1016/j.neubiorev.2022.104965 , 10.1001/jamanetworkopen.2023.12443 etc.

The statement “These findings suggest that while CYP2C19 polymorphisms may affect the metabolism of certain antidepressants, they may not affect others. Additional confounding factors should be considered to confirm these observations.” is rather obvious as CYP2C19 polymorphisms will most likely influence the pharmacokinetic and outcomes of antidepressants which are metabolized by CYP2C19. Please rephrase the original statement.

The limitations section is lacking, but there are some serious limitations to be reported (listed below in the comments to methods section). Provide an appropriate section and describe the methodological limitations of this work.

Methods:

The authors state that “Initially, no filters were applied, including the year of publication. However, the scope was later narrowed to articles published within the last decade.”. Please clarify the reason for limiting the time frame of the data search.

I see one serious limitation to the applied methodology. The authors report that they used phrase “Depressive Disorder, Major” in the search string instead of “Major Depressive Disorder”. Now, this becomes a major issue because it means some valid articles might not be included in the paper. I believe a search is needed with the use of a “free text” approach (which should have been conducted from the start) to make sure all papers of interest are included in the review.

Moreover, the authors do not report on the potential publication bias and heterogeneity. If these were not performed this is a serious limitation and it has to be addressed at least in the limitation section.

Conclusion:

Again the phrase “These findings suggest that CYP2C19 polymorphisms may in fluence the metabolism of some antidepressants but not others, highlighting the need to consider other factors that could interfere with metabolism.” is rather obvious as CYP2C19 polymorphisms will most likely influence the pharmacokinetic and outcomes of antidepressants which are metabolized by CYP2C19. Please rephrase the original statement and report which antidepressants pharmacokinetics in particular were found to be influenced by CYP2C19 polymorphisms.

Comments on the Quality of English Language

The text needs to be once more thoroughly revised and spell-checked.

Author Response

We appreciate the suggestions made and addressed each point raised by the reviewers. We believe that these suggestions increased the overall quality of the submitted manuscript.

Therefore, we are resubmitting our revised systematic review entitled "CYP2C19 genetic variants and Major Depressive Disorder: a systematic review.” Changes in the manuscript and our answers to the reviewers’ comments are in blue.

All authors are aware of the resubmission and agree with the responses to the reviewers provided below.

____________________________

Reviewer 1

Comments and Suggestions for Authors

CYP2C19 genetic variants and Major Depressive Disorder: a systematic review

The article “CYP2C19 genetic variants and Major Depressive Disorder: a systematic review” explores and important issue of CYP 2C19 genetic variants and treatment response in MDD. While the topic is certainly worth exploring there are some significant limitations to the methodology of the study.

Abstract: The abstract in generally well-structured and comprehensive. The reporting of review outcomes is very vague. You have found some data indicating that CYP2C19 polymorphisms are linked to some antidepressants plasma concentrations and treatment response in MDD, report that in the abstract.

Answer: Thank you for your valuable suggestion. We have made the necessary adjustments to the abstract and hope it has become more precise and complete. The following are the changes we made:

“Abstract: Major depressive disorder (MDD) affects over 300 million people globally and has a multifactorial etiology. The CYP2C19 enzyme, involved in metabolizing certain antidepressants, can influence treatment response. Following the PRISMA protocol and PECOS strategy, this systematic review assessed the variation in common CYP2C19 gene variants' frequencies across populations with MDD, evaluating their impact on clinical characteristics and treatment response. We comprehensively searched five databases, identifying 240 articles, of which only nine met our inclusion criteria within the last decade. Except for one study that achieved 74.28% of STROPS items, the rest met at least 75% of GRIPS and STROPS guidelines for quality and bias risk assessment. The CYP2C19's *1 allele, the *1/*1 genotype, and the NM phenotype, considered as references, were generally more frequent. Other CYP2C19 polymorphism frequencies exhibit significant variability across different populations. Some studies associated variants with MDD development, a more extended history of depression, prolonged depressive episodes, and symptom severity, while others reported no such association. Some studies confirmed variants' effects on escitalopram and citalopram metabolism but not that of other drugs, such as sertraline, venlafaxine, and bupropion. Treatment tolerability and symptom improvement also varied between studies. Despite some common findings, inconsistencies highlight the need for further research to clarify the role of these polymorphisms in MDD and optimize treatment strategies.”

Key words: there is an error which is repeated in many times in the manuscript, with the widely used term “Major Depressive Disorder” being written in an uncommon manner “Depressive Disorder, Major”. This need to be corrected throughout the text.

Answer:   Thank you for your valuable feedback. We understand your concern regarding the use of the term "Depressive Disorder, Major" in the manuscript. We would like to clarify that this terminology is based on the Medical Subject Headings (MeSH) vocabulary thesaurus, which is widely used for indexing and cataloging biomedical information in databases such as PubMed. According to MeSH, "Depressive Disorder, Major" is the standardized term for this condition, and we have adhered to this convention in the manuscript's keywords to facilitate finding our manuscript and ensure consistency with recognized indexing systems as well as in our search terms as it's a standardized form to search in different databases. We hope this explanation addresses your concern, but we are happy to make any necessary adjustments should this still pose an issue.

Introduction:

The first paragraph is inconsistent in naming some but not all diagnoses listed in the “Depressive Disorders” category of DSM-5. I see no need to list all types of depressive disorders in this part of the text, especially given that the article refers largely to MDD. Instead, I suggest replacing this part of the text with the later fragment explaining the term “Major depressive disorder” by listing its diagnostic criteria. This particular fragment needs to be rewritten with caution, as some symptoms are not properly described (i.e. “reduced physical movement” instead of „Psychomotor changes” or “sense of uselessness” instead of “sense of worthlessness”). If none of the authors works clinically, I believe it would be best if You would consult a clinician (psychiatrist) prior to providing revisions to the text.

Answer: Thank you for your comments. We talked with our research group's psychologist, and we agree that listing all types of disorders is unnecessary, as this could make the text tiresome for the reader. On the other hand, to clarify matters further, we altered the sentence and included the DSM-V reference in case the reader is interested in delving deeper into the topic, as seen below.

“The term depression generally refers to any depressive disorder. Nonetheless, according to the DSM-V (Diagnostic and Statistical Manual of Mental Disorders, Fifth Edition), these disorders are divided into major depressive disorder (MDD, also known as major depression), persistent depressive disorder, disruptive mood dysregulation disorder, among others [1, DSM V https://doi.org/10.1007/978-981-32-9271-0_9].”

And

“The etiology of MDD is believed to be multifactorial, including biological, genetic, environmental, and psychosocial factors [3]. The DSM-V defines depression as the presence of at least five symptoms that persist for at least two weeks, one of which is depressed mood or loss of interest or pleasure. Symptoms include depressed mood most of the day, markedly reduced interest or pleasure most of the day, weight loss or gain, decreased or increased appetite, insomnia or hypersomnia, agitation or psychomotor retardation, feelings of worthlessness or excessive guilt, fatigue or loss of energy, decreased ability to think or concentrate, recurrent thoughts of death, and suicidal ideation [3,4, DSM-V].”

We hope this makes it more straightforward, but if you still feel it necessary, please let us know so we can complement this information. Regarding signs and symptoms, we have revised the paragraph and believe it is now adequate. We have added another reference to confirm the correct terms. We sincerely hope that this is in line with what you suggested.

The authors state that “The CYP2C19 enzyme helps metabolize a range of clinically utilized medicines, including approximately 15.5% of FDA-approved and routinely prescribed psychiatric drugs, such as citalopram and sertraline, as well as endogenous substances, such as melatonin and progesterone [6,13]”. This work has limitations but it can be of relevance to clinicians. Provide a list or a table of antidepressants which are metabolized by CYP2C19 to a potentially significant extent. Here are potential sources for such a table, but feel free to use any others if You have more comprehensive ones (doi: 10.1080/17425255.2024.2352468; 10.1001/jamapsychiatry.2020.3643).

Answer: Thank you for your insightful comments and suggestions. The role of the CYP2C19 enzyme in metabolizing various drugs, as well as the impact of its genetic variants, has been extensively documented in the literature. Since our systematic review did not focus on this specific aspect, we did not feel it was necessary to include a table listing the antidepressants metabolized by CYP2C19.

That said, we appreciate your suggestion and have added relevant references within this section for readers who may wish to explore the topic further, including one of the references you kindly provided.

“The CYP2C19 enzyme helps metabolize a range of clinically utilized medicines, as reported extensively in the literature [https://doi.org/10.1002/cpt.2903, https://pubmed.ncbi.nlm.nih.gov/32602114/, 10.1001/jamapsychiatry.2020.3643, https://www.pharmgkb.org/gene/PA124/variantAnnotation], including approximately 15.5% of FDA-approved and routinely prescribed psychiatric drugs, such as citalopram and sertraline, as well as endogenous substances, such as melatonin and progesterone [6,13]. “

We hope this addition clarifies the context for readers interested in the clinical relevance of CYP2C19-related drug metabolism. Thank you once again for your constructive feedback.

Results:

Table 1 is well prepared, yet the abbreviations used in the table should be explained in the footnotes. Please provide them.

Table 2 is well prepared, yet the abbreviations used in the table should be explained in the footnotes. Please provide them.

Answer: We added the description of the nucleotides' abbreviations for Tables 1 and 2 in their footnotes.

The flow chart is comprehensible and well organized.

Answer: Thank you for your comment. The flow chart is from the PRISMA protocol.

The description “American” comes up several times but it is not precise. Please specify whether You mean North/South American or both by “American”.

Answer: Thank you for your comment. As suggested, we have changed the first paragraph of item 2.2 to clarify this issue and added specifications in the text to what population it was referring to. We hope this clarifies it.

“As shown in Figure 2, most studies were conducted on the American (United States, Canada, and Trinidad and Tobago) and European (Germany, Serbia, and Poland) continents, followed by transcontinental Turkey and Asia (China). Regarding demographic data, most research was conducted with adults over 18, and women were more prevalent in the Major Depressive Disorder (MDD) groups.”

All figures are of very low quality. Please provide Figures with higher resolution.

Answer: Thank you for your suggestion. We added higher-resolution figures.

In the section “2.2” the abbreviation “MDD” is all of a sudden introduced while it should ne provided in the part of the text where the phrase “Major Depressive Disorder” appears first.

Answer: Thank you very much for your observation, but we are unsure if we understand the problem exposed. The meaning of the acronym MDD is cited in the abstract, the first paragraph of the introduction, at the end of the first paragraph of item 3.2 (when it's first introduced in that section), the first paragraph of the discussion, and the tables and figures.

Table 3 has misspells and lacks appropriate use of MDD abbreviations, there are some hyperlinks in the Table instead of a proper citation. Please correct this table once more and do it thoroughly.

Answer: Thank you for your valuable feedback. We checked the spelling and abbreviated terms. We also removed or exchanged the hyperlinks in the table for references.

Regarding Table 3, why do the authors report the assessed drugs for some studies but do not for others? Whenever the data is available the drugs evaluated in a particular study should be listed in the table.

Answer: We appreciate the opportunity to clarify a few points. Not all studies included in our review assessed plasma levels of drugs and metabolites. As our focus was on the frequency variation of CYP2C19 genetic variants in different populations with MDD, there was significant heterogeneity in the observational and interventional studies in this regard. Unfortunately, this variability made it impossible to report the specific drugs for all studies in Table 3.

Some studies focused on susceptibility to developing MDD, while others measured plasma concentrations of antidepressants and metabolites or assessed treatment response and tolerability. We hope this explanation clarifies the differences in reported data across studies.

Discussion:

Please specify whether You mean North/South American or both by “American”.

Answer: Thank you for your comment. As suggested, we have changed the first paragraph of item 3.1 to clarify this issue and added specifications in the text to what population it was referring to.

“Most studies in European, transcontinental Turkey (Asia/Europe) and American (North and Caribbean) populations found a higher NM (normal metabolizer) phenotype (*1/*1 genotype) frequency among Major Depressive Disorder (MDD) patients, with IM (intermediate metabolizer) and RM (rapid metabolizer) alternating as the second most frequent and PM consistently being the least common (Figure 5, Tables 3 and S3).”

and

“In contrast, their second most frequent phenotype was RM (31.34%). Interestingly, these studies' least frequent phenotypes diverged, with the Canadian being PM (2.83%) and the United States being UM (2.99%).”

We hope this clarifies it.

The “BDI-IA” needs to be explained upon it’s first occurrence in the text.

Answer: We are grateful for your point. We have corrected this in the footnote of Table 3, where the acronym is mentioned for the first time, and added the meaning of the acronym in its first appearance in the discussion.

If You wish to use abbreviation for the word “citalopram” introduce it upon it’s first occurrence in the text, same goes for “sertraline”, “desmethylsertraline” etc.

Answer: We appreciate your concern and thank you for your suggestions. We have inserted the meaning of the acronyms at the bottom of Table 3 and added several drug abbreviations, cited for the first time in the text, to the table. We have also taken advantage of this editing to improve the table's footer further concerning other acronyms mentioned.

The discussion is quite long but it is a description of the results rather than a discussion. Discuss Your results with the relevant literature i.e. 10.1001/jamapsychiatry.2020.3643, 10.1111/bcp.15762 , 10.1016/j.neubiorev.2022.104965 , 10.1001/jamanetworkopen.2023.12443 etc.

Answer: Thank you very much for your suggestion. We have inserted a few more references in the discussion accordingly. Below, we highlight the passage where one of the references was inserted:

“In another study of 227 patients with depressive disorders, Hall-Flavin et al. [39] analyzed various CYP2C19 polymorphisms (*1, *2, *3, *4, *5, *6, *7, and *8) and evaluated differences between patients with treatment guided and not guided by pharmacogenetic tests. They found that patients with pharmacogenetically-guided treatment showed a significant reduction in depressive symptoms on the HAMD-17 scale compared to the non-guided group. Arnone et al. [https://doi.org/10.1016/j.neubiorev.2022.104965] reported similar results in a meta-analysis, finding that when pharmacogenomic tests were employed to tailor the treatment of depression, they were more effective than the usual methods according to the improvement assessed in MDD patients.”

Furthermore, we have inserted new paragraphs at the end of items 4.2 and 4.3 of the discussion, summarizing the results found. We hope this clarifies and fills the gap. Below are the inserted passages:

Item 4.2:

“Not all selected studies assessed the CYP2C19 gene polymorphisms' association with MDD clinical characteristics, revealing heterogeneity in regard to which outcomes each study evaluated. The ones that did, analyzed the possibility of these polymorphisms associated with MDD development, as well as with a more extended history of depression, longer duration of the current depressive episode, and symptom severity [30,31]. Some confirmed these associations [44], while others reported no such association [21, 29,34,43].

The plasma concentrations of the antidepressant drugs and their metabolites were also not evaluated in all selected articles. Uckun et al., for example, found that the CYP2C19*2 polymorphism impacted desmethylcitalopram plasma concentrations [26]. Other studies agree with these results, showing the impact of CYP2C19 gene polymorphisms on the metabolization of escitalopram and citalopram [17,25,45]. Alternately, other of the selected studies found no association between CYP2C19 gene polymorphisms and the sertraline/desmetilsertraline, venlafaxine/O-desmetilvenlafaxine, and bupropion metabolization, as demonstrated by their unchanged plasma levels [27,28,33].

These dissimilarities among the selected studies demonstrate inconsistencies regarding the CYP2C19 gene polymorphisms' association with MDD clinical characteristics and certain drug metabolizations, making it essential for studies to analyze confounding factors to confirm associations in varying contexts. Following study-type guidelines could also help standardize methods to regulate variables and compare results, thus improving the understanding of factors affecting the study's outcomes.”

Item 4.3:

“Regarding the response to treatment, among the selected studies that evaluated this outcome, patients with SM phenotype also improved their conditions after pharmacotherapy [29,30], even though those with NM phenotype had a more pronounced improvement [30]. This relationship has been confirmed by several authors [23,39,40, Arnone et al.]. However, contrasting results have also been found, showing no association between CYP2C19 gene polymorphisms and a better response to treatment [17,21,45,46,47,48,50].

Whereas with regard to treatment tolerability, determined by the occurrence of side effects and adverse reactions, among the reviewed results, MDD patients with the SM phenotype had more adverse reactions and side effects than patients with the NM phenotype [30,32]. Other studies confirmed [45] or found no differences [49] regarding the lower tolerability to treatment in the MS group. The same was observed in terms of sexual function improvement following therapy, with one study finding no difference in improvement between the phenotype groups [30] and another indicating that the SM phenotype groups improved more than the other groups [32].

These dissimilarities in results reiterate the importance of standardizing treatment and following study-type guidelines to regulate and better understand the elements that can generate disparate results of different studies.”

The statement “These findings suggest that while CYP2C19 polymorphisms may affect the metabolism of certain antidepressants, they may not affect others. Additional confounding factors should be considered to confirm these observations.” is rather obvious as CYP2C19 polymorphisms will most likely influence the pharmacokinetic and outcomes of antidepressants which are metabolized by CYP2C19. Please rephrase the original statement.

Answer: Thank you for your comments. We've added more references to the discussion to complement your suggestions. We hope this will make it more appropriate and further improve the quality of the article. Here is the amended excerpt:

“These findings suggest that while CYP2C19 polymorphisms may affect the metabolism of certain antidepressants, they may not affect others [https://doi.org/10.1002/cpt.2903, https://pubmed.ncbi.nlm.nih.gov/32602114/, 10.1001/jamapsychiatry.2020.3643, https://www.pharmgkb.org/gene/PA124/variantAnnotation]. Nevertheless, in order to clarify this point, it is crucial to evaluate confounding factors that may interfere with this metabolization since they generally cannot be controlled entirely [https://doi.org/10.1001/jamapsychiatry.2020.3643], such as the patient’s consumption of grapefruit juice [https://doi.org/10.1080/17425255.2024.2352468].”

The limitations section is lacking, but there are some serious limitations to be reported (listed below in the comments to methods section). Provide an appropriate section and describe the methodological limitations of this work.

Answer: Thank you for your valuable suggestion; it will further improve the quality of our work.

To remedy this shortcoming, we have added a paragraph in the conclusion stating the review's limitations.

“This review had some limitations. Although all the studies reported allelic, genotypic, and phenotypic frequencies —our primary focus— the populations studied varied in particular characteristics and the number of participants; for instance, some had a small sample size. These differing respective characteristics and sample sizes might limit the generalizability of this review's results. Furthermore, there was also heterogeneity in the outcomes assessed, where some investigated the plasma concentrations of drugs and their metabolites, for example, and others investigated the response and tolerability to treatment. Lastly, not all studies analyzed the same CYP2C19 polymorphisms, making it hard to comprehensively analyze all the phenotypes and genotypes.”

In addition, in the methodology, results, and discussion, we included the assessment of the quality of the articles selected using the Genetic Risk Prediction Studies (GRIPS) guideline and the STrengthening the Reporting of Pharmacogenetic Studies (STROPS) Guideline. We hope this will make evaluating the review's limitations more straightforward.

The authors state that “Initially, no filters were applied, including the year of publication. However, the scope was later narrowed to articles published within the last decade.”. Please clarify the reason for limiting the time frame of the data search.

Answer: Thank you for your comment. The PRISMA protocol allows for limiting the period for the review. After the initial search, we chose to select more recent studies to analyze the 'state of the art' as many publications passed the PECOS strategy while investigating different outcomes allowed us this choice.

I see one serious limitation to the applied methodology. The authors report that they used phrase “Depressive Disorder, Major” in the search string instead of “Major Depressive Disorder”. Now, this becomes a major issue because it means some valid articles might not be included in the paper. I believe a search is needed with the use of a “free text” approach (which should have been conducted from the start) to make sure all papers of interest are included in the review.

Answer: Thank you for your valuable feedback. We followed the Preferred Reporting Items for Systematic Reviews and Meta-Analyses (Prisma) guidelines for systematic reviews to ensure the rigor and transparency of our methodology and Population, Exposure, Comparison, Outcome, and Study type (PECOS) strategy to delimit our search, approved under PROSPERO number CRD42024525997. From my understanding, the "free text" approach is normally used for narrative and integrative reviews where the results don't require repeatability of the search results.

While we understand your concern regarding using the term "Depressive Disorder, Major" in our search, this term is the standard for this disorder per the Medical Subject Headings (MeSH) vocabulary thesaurus. We hope this clarifies your concern. We will be happy to make any necessary adjustments that uphold the PRISMA protocol and PECOS strategy should this still pose an issue.

Moreover, the authors do not report on the potential publication bias and heterogeneity. If these were not performed this is a serious limitation and it has to be addressed at least in the limitation section.

Answer: Thank you for your valuable suggestion; we have added in the methodology, results, and discussion the assessment of the quality and bias risk of the selected articles using the Genetic Risk Prediction Studies (GRIPS) guideline and the STrengthening the Reporting of Pharmacogenetic Studies (STROPS) Guideline. As for the heterogeneity in the publications, it is not a problem or limitation as our focus was on the frequency variation of CYP2C19 genetic variants in different populations with MDD, and all selected articles passed the PECOS strategy even though they were investigating different outcomes.

Again the phrase “These findings suggest that CYP2C19 polymorphisms may in fluence the metabolism of some antidepressants but not others, highlighting the need to consider other factors that could interfere with metabolism.” is rather obvious as CYP2C19 polymorphisms will most likely influence the pharmacokinetic and outcomes of antidepressants which are metabolized by CYP2C19. Please rephrase the original statement and report which antidepressants pharmacokinetics in particular were found to be influenced by CYP2C19 polymorphisms.

Answer: We modified the paragraph as follows:

“These findings confirm that while CYP2C19 polymorphisms may affect the metabolism of certain antidepressants, they may not affect others [https://doi.org/10.1002/cpt.2903, https://pubmed.ncbi.nlm.nih.gov/32602114/, 10.1001/jamapsychiatry.2020.3643, https://www.pharmgkb.org/gene/PA124/variantAnnotation]. Nevertheless, in order to clarify this point, it is crucial to evaluate confounding factors that may interfere with this metabolization since they generally cannot be controlled entirely [https://doi.org/10.1001/jamapsychiatry.2020.3643], such as the patient’s consumption of grapefruit juice [https://doi.org/10.1080/17425255.2024.2352468].”

Comments on the Quality of English Language

The text needs to be once more thoroughly revised and spell-checked.

Answer: Thank you very much for your comments. The manuscript has been written and reviewed by native English-speaking authors, and we believe the language is clear after careful revision. However, we greatly value your input, so if there are specific areas you feel could be further improved, please let us know, and we will be happy to make any necessary adjustments.

Reviewer 2 Report

Comments and Suggestions for Authors

The present systematic review, “CYP2C19 genetic variants and Major Depressive Disorder: a systematic review,” is a significant contribution that identifies the frequency fluctuation of the  CYP2C19 genetic variants in different populations with major depressive disorder. It also describes how these polymorphisms influence major depressive disorder clinical characteristics and the response to antidepressants, underscoring the importance of this research in the field of psychiatry and pharmacogenomics.

The data presented herein are not only comprehensive but also valuable and engaging, providing a thorough understanding of the topic at hand.

The entire manuscript is very well-written and adequately illustrated. It is accompanied by adequately summarized tabular data in the main manuscript text and presented supplementary data. The extracted data are adequately discussed and summarized correctly in the concluding section of the manuscript.

The acceptance of the manuscript in its current form is suggested.

Author Response

We appreciate the suggestions made and addressed each point raised by the reviewers. We believe that these suggestions increased the overall quality of the submitted manuscript. 

Therefore, we are resubmitting our revised systematic review entitled " CYP2C19 genetic variants and Major Depressive Disorder: a systematic review.” Changes in the manuscript and our answers to the reviewers’ comments are in blue. 

____________________________ 

Reviewer 2 

Comments and Suggestions for Authors 

The present systematic review, “CYP2C19 genetic variants and Major Depressive Disorder: a systematic review,” is a significant contribution that identifies the frequency fluctuation of the CYP2C19 genetic variants in different populations with major depressive disorder. It also describes how these polymorphisms influence major depressive disorder clinical characteristics and the response to antidepressants, underscoring the importance of this research in the field of psychiatry and pharmacogenomics. 

The data presented herein are not only comprehensive but also valuable and engaging, providing a thorough understanding of the topic at hand. 

The entire manuscript is very well-written and adequately illustrated. It is accompanied by adequately summarized tabular data in the main manuscript text and presented supplementary data. The extracted data are adequately discussed and summarized correctly in the concluding section of the manuscript. 

The acceptance of the manuscript in its current form is suggested. 

Answer: Thank you very much for your kind and encouraging feedback on our manuscript. We are truly grateful for your recognition of the significance of our work and for your positive assessment of its clarity and thoroughness. Your remarks regarding the data’s comprehensiveness and value, as well as the presentation and discussion, are deeply appreciated. 

We are delighted that you found the manuscript to be well-written and engaging, and we sincerely thank you for recommending its acceptance. 

All authors are aware of the resubmission and agree with the responses to the reviewers provided below. 

Reviewer 3 Report

Comments and Suggestions for Authors

The manuscript is a systematic review of research on CYP2C19 genetic polymorphisms in patients with depression. Genetic polymorphisms occur at different frequencies in different populations, so the efficacy of different antidepressants may vary according to ethnicity. Therefore, such studies are of direct interest to clinical professionals. The authors have made very good and clear tables. 

Perhaps the authors should replace the graphs in chapter 2.3 and 2.4. with a table showing the mean frequency, spread and minimum maximum value for all studies.

The authors should also add how their review differs from other reviews on this topic, how it is new and unique. For example, compared to this review:

https://www.ncbi.nlm.nih.gov/pmc/articles/PMC7702196/

Author Response

We appreciate the suggestions made and addressed each point raised by the reviewers. We believe that these suggestions increased the overall quality of the submitted manuscript.

Therefore, we are resubmitting our revised systematic review entitled "CYP2C19 genetic variants and Major Depressive Disorder: a systematic review.” Changes in the manuscript and our answers to the reviewers’ comments are in blue.

All authors are aware of the resubmission and agree with the responses to the reviewers provided below.

-----

Comments and Suggestions for Authors

The manuscript is a systematic review of research on CYP2C19 genetic polymorphisms in patients with depression. Genetic polymorphisms occur at different frequencies in different populations, so the efficacy of different antidepressants may vary according to ethnicity. Therefore, such studies are of direct interest to clinical professionals.

The relevance of this work is due to the fact that the CYP2C19 gene has many genetic variations. The presence of certain genetic variants may influence the treatment response. There are many articles on this topic, but there is only one systematic review. The authors should add how their review differs from other reviews on this topic, how it is new and unique. For example, compared to this review: https://www.ncbi.nlm.nih.gov/pmc/articles/PMC7702196/.

Answer: Thank you for pointing out the systematic review and meta-analysis by Jukic et al. (2020), which explores the impact of genetically determined CYP2C19 and CYP2D6 metabolism on antipsychotic and antidepressant exposure. Although our topics are similar, our focus is distinct. Our systematic review investigated CYP2C19 genetic variants' frequency variation in different populations, specifically with major depressive disorder (MDD), and, when available in the selected study, how these variants influence both clinical characteristics of MDD and response to antidepressants. Therefore, while both reviews discuss the role of genetic polymorphisms, our work focuses on population genetics and MDD-specific outcomes, offering a complementary yet different perspective.

We modified the last paragraph of Introduction to reflect your suggestion:

“Many studies have sought to gain insight into how polymorphisms of the CYP2C19 gene affect treatment outcomes, adverse effects, and their frequency in different populations, trying to identify gaps in this understanding [10.1001/jamapsychiatry.2020.3643; 10.3389/fphar.2024.1326776; 10.1001/jamanetworkopen.2023.12443; 10.1111/bcp.15762]. These findings have led to Pharmacogene Variation Consortium (PharmVar) catalogs star (*) allele nomenclature for the polymorphic human CYP2C19 gene and a reviewed consensus on pharmacogenomic testing and their effectiveness in psychiatry, explaining how CYP2C19 genetic variation impacts the metabolism of many drugs and informing medication selection and dosing of several commonly-used antidepressant and antipsychotic medications [10.1055/a-1288-1061; https://doi.org/10.1002/cpt.1973, 10.1016/j.neubiorev.2022.104965]. In light of the global prevalence of MDD, the variability in treatment response between individuals, and the CYP2C19 enzyme's role in metabolizing antidepressant drugs, this systematic review aimed to determine the CYP2C19 genetic variants' frequency variation in different populations with major depressive disorder and to understand how these polymorphisms influence MDD clinical characteristics and the response to antidepressants.”

The methodology follows the PRISMA criteria for systematic reviews.

Answer: Thank you for your observation. We would like to confirm that we followed the PRISMA guidelines for systematic reviews to ensure the rigor and transparency of our methodology.

In paragraphs 3.2 and 3.3, the authors should add a concluding sentence summarizing the information given in these parts.

Answer: Thank you for your suggestion. As requested, we have inserted new paragraphs at the end of items 4.2 and 4.3 of the discussion, summarizing the results found. We hope this clarifies and fills the gap. Below are the inserted passages:

Item 4.2:

“Not all selected studies assessed the CYP2C19 gene polymorphisms' association with MDD clinical characteristics, revealing heterogeneity in regard to which outcomes each study evaluated. The ones that did, analyzed the possibility of these polymorphisms associated with MDD development, as well as with a more extended history of depression, longer duration of the current depressive episode, and symptom severity [30,31]. Some confirmed these associations [44], while others reported no such association [21, 29,34,43].

The plasma concentrations of the antidepressant drugs and their metabolites were also not evaluated in all selected articles. Uckun et al., for example, found that the CYP2C19*2 polymorphism impacted desmethylcitalopram plasma concentrations [26]. Other studies agree with these results, showing the impact of CYP2C19 gene polymorphisms on the metabolization of escitalopram and citalopram [17,25,45]. Alternately, other of the selected studies found no association between CYP2C19 gene polymorphisms and the sertraline/desmetilsertraline, venlafaxine/O-desmetilvenlafaxine, and bupropion metabolization, as demonstrated by their unchanged plasma levels [27,28,33].

These dissimilarities among the selected studies demonstrate inconsistencies regarding the CYP2C19 gene polymorphisms' association with MDD clinical characteristics and certain drug metabolizations, making it essential for studies to analyze confounding factors to confirm associations in varying contexts. Following study-type guidelines could also help standardize methods to regulate variables and compare results, thus improving the understanding of factors affecting the study's outcomes.”

Item 4.3:

“Regarding the response to treatment, among the selected studies that evaluated this outcome, patients with SM phenotype also improved their conditions after pharmacotherapy [29,30], even though those with NM phenotype had a more pronounced improvement [30]. This relationship has been confirmed by several authors [23,39,40, Arnone et al.]. However, contrasting results have also been found, showing no association between CYP2C19 gene polymorphisms and a better response to treatment [17,21,45,46,47,48,50].

Whereas with regard to treatment tolerability, determined by the occurrence of side effects and adverse reactions, among the reviewed results, MDD patients with the SM phenotype had more adverse reactions and side effects than patients with the NM phenotype [30,32]. Other studies confirmed [45] or found no differences [49] regarding the lower tolerability to treatment in the MS group. The same was observed in terms of sexual function improvement following therapy, with one study finding no difference in improvement between the phenotype groups [30] and another indicating that the SM phenotype groups improved more than the other groups [32].

These dissimilarities in results reiterate the importance of standardizing treatment and following study-type guidelines to regulate and better understand the elements that can generate disparate results of different studies.”

The conclusions are correct and reflect the essence of the review. The authors report that data on the effects of polymorphisms on antidepressant metabolism and clinical characteristics are inconsistent. Perhaps the authors should suggest what caused such inconsistencies. For example, because of different methodology or the small number of people involved in the studies.

Answer: Thank you. Your comment was very valuable to us. To remedy this shortcoming, we have added a paragraph to the conclusion stating the limitations of the review. We believe that this paragraph (as seen below) will address your suggestions and further improve the quality of our work.

“This review had some limitations. Although all the studies reported allelic, genotypic, and phenotypic frequencies —our primary focus— the populations studied varied in particular characteristics and the number of participants; for instance, some had a small sample size. These differing respective characteristics and sample sizes might limit the generalizability of this review's results. Furthermore, there was also heterogeneity in the outcomes assessed, where some investigated the plasma concentrations of drugs and their metabolites, for example, and others investigated the response and tolerability to treatment. Lastly, not all studies analyzed the same CYP2C19 polymorphisms, making it hard to comprehensively analyze all the phenotypes and genotypes.”

Furthermore, in the methodology, results, and discussion, we included our analysis of the selected articles using the Genetic Risk Prediction Studies (GRIPS) and Strengthening the Reporting of Pharmacogenetic Studies (STROPS) guidelines for quality and bias risk assessment.

The authors have made very good and clear tables. However, the results in Table 3 should be written more briefly.

Answer: Thank you for your suggestion. Following your recommendation, we have summarized the information as much as possible so that no critical information is lost. We sincerely hope that it is more objective this way.

Perhaps the authors should replace the graphs in chapter 2.3 and 2.4. with a table showing the mean frequency, spread and minimum maximum value for all studies.

Answer: Thank you very much for your feedback. The data illustrated in the graphs in chapters 3.3 and 3.4 (Figures 3, 4, and 5) are also described in Tables S2 and S3 of the supplementary material to allow the readers a more in-depth comprehension of each study's data. We added the graphs to the manuscript because we believe it would be more attractive and accessible for readers to view the results. However, these tables in the supplementary material do not report mean, maximum, and minimum values.

Unfortunately, pooling the frequency results of each study to calculate a frequency mean is unfeasible as these studies have different populations. We believe you might be referring to an analysis done in systematic reviews with meta-analysis that analyzes risk by pooling odds ratios (ORs), which have confidence intervals and are independent of population differences. However, applying this method to frequencies would not be appropriate, as they inherently vary by population. In our systematic review, we chose not to pursue this approach because many of the studies included were not risk studies, and those that were, some presented ORs in different formats—some provided raw ORs, while others were adjusted for different confounders, such as age, which can't be pooled together. We understand that it would have been a fantastic addition, and in the future, we will undoubtedly consider incorporating it in future meta-analyses. Once again, we appreciate your insightful suggestion.

Round 2

Reviewer 1 Report

Comments and Suggestions for Authors

The authors have appropriately addressed the vast majority of my concerns.

The use of English in this manuscript is now appropriate.

Regarding the search strategy. I certainly understand the use of MeSH term in the search strategy. It’s worth noting that COCHRANE recommends using also free text search https://training.cochrane.org/handbook/current/chapter-04 , such an approach is also recommended in PRESS Guideline Statement 10.1016/j.jclinepi.2016.01.021 and considered valid by PRISMA-S group10.1186/s13643-020-01542-z. Now because, authors explicitly state that their search was performed with the of Mesh term only, I don’t think there is a need to rerun the search. Please, keep these above mentioned works in mind for Your future research, so that Your efforts can result in the work of highest quality.

My last concern is that the largely limited scope of included data (limits on the data of publication and lack of use of the free text search), which authors mention themselves in the method section “due to the large volume of publications, the scope was later narrowed to articles published within the last decade” this work provides a summary of only some of the data available in that area. I suggest the authors state that in the limitations section.

Author Response

Comments and Suggestions for Authors 

The authors have appropriately addressed the vast majority of my concerns. 

Answer: Thank you for your comment and we're happy to hear it. 

The use of English in this manuscript is now appropriate. 

Answer: Thank you for your comment and we're glad we were able to adapt it as you suggested. 

Regarding the search strategy. I certainly understand the use of MeSH term in the search strategy. It’s worth noting that COCHRANE recommends using also free text search https://training.cochrane.org/handbook/current/chapter-04 , such an approach is also recommended in PRESS Guideline Statement 10.1016/j.jclinepi.2016.01.021 and considered valid by PRISMA-S group10.1186/s13643-020-01542-z. Now because, authors explicitly state that their search was performed with the of Mesh term only, I don’t think there is a need to rerun the search. Please, keep these above mentioned works in mind for Your future research, so that Your efforts can result in the work of highest quality. 

Answer: Thank you for your clarification. Your suggestion will be valuable to our research group. Rest assured that we will keep the works mentioned in mind for our subsequent publications. 

My last concern is that the largely limited scope of included data (limits on the data of publication and lack of use of the free text search), which authors mention themselves in the method section “due to the large volume of publications, the scope was later narrowed to articles published within the last decade” this work provides a summary of only some of the data available in that area. I suggest the authors state that in the limitations section. 

Answer: Thank you for your comments. We added an excerpt in section 3.1 to clarify this issue: 

“Initially, 240 articles were identified across the four databases searched. After removing duplicates, 172 studies remained for the title and abstract analysis, taking into account the aspects defined in our PECOS strategy. Of these, 45 articles persisted for full-text review. Finally, after applying the inclusion and exclusion criteria, nine articles were included in this systematic review (Figure 1, Table 3). Our choice of the limited publication period (2014-2024) favored the selection of more recent information on the research subject. As the fields of pharmacology, genetic screening, and gene-disease modeling are constantly updated, this approach gives insights into the latest publications. The excluded articles and their reason for exclusion are described in Table S1.”  

Dear reviewer, we understand your concern, but we are also of the understanding that it is also essential for the reader to keep up with the most recent information and latest publications on the research subject, especially as the fields of pharmacology and genetic screening and modeling are constantly updated. Therefore, we sincerely believe que these delimitations might enrich rather than limit our work.  

Regarding the use of free-text search, our primary concern was ensuring the reproducibility of our systematic review. We reviewed your recommended Chapter 4: Searching for and selecting studies | Cochrane Training and acknowledge that, indeed, its recommended search methods combine controlled vocabulary (e.g., MeSH terms) with free-text terms to enhance sensitivity and reduce the risk of missing relevant studies and is also recognized as a good practice by PRISMA-S extension. Nevertheless, we adhered to PRISMA guidelines that prioritize transparency in reporting search strategies over prescribing specific methodologies and allow for flexibility in choosing search strategies as long as the methods are well-documented and thorough. In our case, using MeSH terms enabled a systematic and reproducible approach, and our search successfully identified a substantial number of relevant and well-documented studies; therefore, it should still meet the standard requirements for a high-quality review. Thanks again for the suggestion.  We will incorporate a free-text search method combined with controlled vocabulary for our future research, as it could further enhance the comprehensiveness of our strategy. We sincerely hope this question has been clarified and we have provided you with the necessary information.